



# Applicability of the VisiSize D30 shadowgraph system for cloud microphysical measurements

Jakub L. Nowak[1,*], Moein Mohammadi[1,*], and Szymon P. Malinowski[1]

[1]Institute of Geophysics, Faculty of Physics, University of Warsaw, Warsaw, 02-293, Poland
[*]These authors contributed equally to this work.

**Correspondence:** Jakub L. Nowak (jakub.nowak@fuw.edu.pl)

**Abstract.** The commercial shadowgraph system, Oxford Lasers VisiSize D30, originally designed to characterize industrial and agricultural sprays, was tested with respect to the application for measuring cloud microphysical properties, such as droplet size distribution and number concentration. Laboratory experiment with a dense stream of poly-disperse cloud-like droplets indicated strong dependence of the depth of field, thus also sample volume, on particle size. This relationship was determined and

a suitable correction method was developed to improve estimations of droplet number concentration and size distribution. Spatial homogeneity of detection probability inside the sample volume and minimum droplet diameter providing uniform detection were examined. The second experiment with mono-disperse droplets produced by Flow-Focusing Monosized Aerosol Generator (FMAG) verified sizing accuracy and demonstrated reasonable agreement between the instruments. Effects of collisions and evaporation of droplets produced by FMAG were observed. Finally, the instrument was applied to sample atmospheric clouds

at a ground-based mountain observatory and performed reliably during 3 week long field experiment. Based on the laboratory and field tests, recommendations concerning the use of the instrument for cloud droplet measurements were formulated.

## 1   Introduction

Atmospheric clouds predominantly consist of water droplets. Cloud droplet number concentration (DNC) and size distribution (DSD) constitute the key parameters for quantitative microphysical description of clouds and attract enormous attention in con-

temporary atmospheric sciences, mostly due to their crucial importance for cloud lifetime, radiative effects and rain formation (Devenish et al., 2012).

There exist two general approaches to measuring cloud microphysical properties: in situ sampling from airborne platforms or at ground-base stations and remote sensing which involves applying inverse retrieval techniques to data collected by satellites, radars and radiometers. Researchers employing both strategies tackle with intrinsic difficulties. However, in situ measurements

are considered fundamental, as they offer instrumental access to individual droplets within a sampling volume. The results obtained in situ are then used to derive and validate inversion routines to be used in remote sensing applications.

Among in situ techniques, one can distinguish two branches differing in sampling style:

- instruments detecting and counting droplets one-by-one but almost continuously in time as they pass through a very small active probe volume provide individual droplet properties and their inter-arrival times,



– instruments capturing images or other spatial representation of droplets inside a larger sampling volume provide individual droplet properties and information on their spatial arrangement.

The first branch is represented by a number of spectrometers using light scattering for droplet detection and sizing, e.g. Forward Scattering Spectrometer Probe (FSSP, e.g. Cooper (1988); Brenguier et al. (1998); Gerber et al. (1999); de Araújo Coelho et al. (2005)), Cloud Droplet Probe (CDP, e.g. McFarquhar et al. (2007); Lance et al. (2010); Lance (2012)), Cloud and Aerosol

Spectrometer (CAS, e.g. Lance (2012); Glen and Brooks (2013); Barone et al. (2019)), Phase Doppler Interferometer (PDI, e.g. Bachalo and Houser (1984); Chuang et al. (2008); Kumar et al. (2019)). In practice, they sample quasi one-dimensional portion of air passing through the region of active sampling. The exact volume sampled in unit time depends on the velocity with respect to the instrument. Therefore, meaningful estimation of DNC requires information on air velocity. Moreover, due to small probe volume cross-section, there exist an upper limit for measurable droplet size. For instance, the range of diameters

of the CDP equals 2-50 µm.

Volumetric methods (second branch) usually do not rely on scattering intensity of individual droplet in sizing and their sampling volumes do not depend on the velocity of the flow but consecutive air samples, collected instantaneously, might be quite distant from each other, in particular when used on fast moving airborne platforms. This class of techniques is represented e.g. by shadowgraphy and holography. Neglecting the effect of imperfect focusing, shadow image constitutes two-dimensional

projection of all the objects onto camera plane and can be rapidly processed to detect particles and obtain relevant statistics. Holograms require extensive processing to digitally reconstruct objects' shape and arrangement in three dimensions (Fugal et al., 2009). Both methods allow to study position, size and shape, thus not only spherical droplets but also e.g. ice crystals.

Holographic systems have been successfully deployed on research aircrafts (HOLODEC, Fugal and Shaw (2009) and HALOHOLO, Schlenczek et al. (2017)), mountain cable cars (HOLOGONDEL, Beck et al. (2017)) and ground-based moun-

tain observatories (HOLIMO, Henneberger et al. (2013)). Typically, those instruments have sample volume (SV) of about 15 $cm^3$, resolution of 6 µm and take a few holograms per second. Shadowgraphy has been used e.g. in Cloud Particle Imager (CPI, Connolly et al. (2007)) - an airborne instrument to observe ice particles and supercooled droplets in the size range 10-2300 µm. Typical SV is much smaller than in holography (about 0.04 $cm^3$), however the frames rate is much higher (400 fps). Moreover, Rydblom and Thörnberg (2016) have designed a system to investigate icing conditions for wind turbines based on

shadow images. Nevertheless, still shadowgraph did not gain common use in cloud measurements despite both its simplicity and many insightful laboratory experiments, e.g. concerning droplet collisions (Bordás et al., 2013; Bewley et al., 2013).

In order to explore in detail the advantages and disadvantages of shadowgraphy for cloud microphysical applications, we use the commercial shadowgraph system (VisiSize D30, Oxford Lasers Ltd., Kashdan et al. (2003, 2004)), originally designed for diagnosis of agricultural and industrial sprays, to measure DNC and DSD in warm clouds. Within the study, two series

of laboratory experiments were performed which aimed at verifying reliability of detection and accuracy of sizing under conditions resembling atmospheric clouds. They were followed by a field experiment targeting real cloud at a mountaintop observatory.

The present paper is structured as follows. Section 2 describes the instrument and the measurement principle. Section 3 provides the analysis of detection reliability and homogeneity affecting results on DNC and DSD, investigated with the use of





poly-disperse water droplets. Section 4 focuses on sizing accuracy studied with the use of mono-disperse droplet population. Based on the experiments, corrections to the standard algorithm implemented in instrument software are suggested. Finally, in section 5 we present selected results obtained during the first application of the instrument in atmospheric clouds. The last section summarizes the findings and discusses the conclusions concerning the further usage of the VisiSize D30 system for cloud research.

## 2 System overview


VisiSize D30 is a complete shadowgraph system manufactured by Oxford Lasers Ltd. (Oxon, United Kingdom) designed to characterize particles in various suspensions. Common industrial applications include among others the characterization of: agricultural sprays, paint sprays, consumer aerosols, fire extinguishers and automotive fuel injectors.

### 2.1 Hardware description

Two main parts of the set-up are infrared diode pulse laser with diffuser and digital camera with suitable lens objective (see Fig. 1). Lens magnification can be changed manually to adjust the resolution and extent of the sample volume to the object of study. For three selected options, instrument calibration was performed by the supplier. Capabilities of the system at those settings are listed in Table 1.

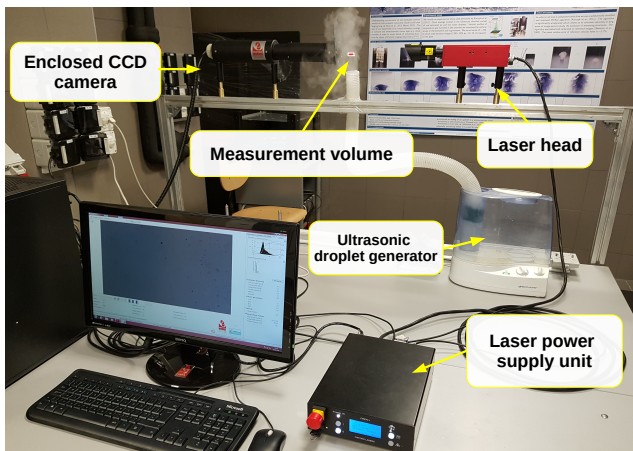

**Figure 1.** Experimental setup for studying detection properties (see sec. 3 for details). Main parts of the VisiSize D30 are: infrared pulse laser with a diffuser (top-right), CCD camera with lens objective enclosed in a water-proof housing (top-left). Water droplets produced with an ultrasonic droplet generator are measured while passing through the sample volume (top-middle) located close to the camera.

The operation of the system is the following. The region of interest is illuminated from behind by diffused (incoherent) 75 expanded laser light beam, then shadow images of droplets are collected at up to 30 frames or pairs of frames per second with a digital camera. The laser and the camera are jointly triggered so that a single laser pulse "freezes" the motion of droplets



**Table 1.** Properties of the VisiSize D30 system for three different lens magnification settings provided by Oxford Lasers Ltd.

| VisiSize D30 specifications | | | |
|---|---|---|---|
| Camera chip [pix x pix] | 1952 x 1112 | | |
| Camera pixel [μm] | 5.5 | | |
| Frame rate [fps] | 30 | | |
| Wavelength [nm] | 808 | | |
| Laser pulse duration [μs] | 0.1 - 5.0 | | |
| Lens setting | x1 | x2 | x4 |
| Magnification | 1.49 | 2.97 | 6.12 |
| Effective pixel size [μm] | 3.69 | 1.85 | 0.90 |
| Resolution [μm] | 6.9 | 3.7 | 2.0 |
| Field of view [mm x mm] | 7.20 x 4.10 | 3.62 x 2.06 | 1.75 x 1.00 |
| Depth of field [mm] | 50.0 | 16.6 | 5.2 |
| Sample volume [$cm^3$] | 1.48 | 0.123 | 0.0092 |
| Volume rate [$cm^3\,s^{-1}$] | 44.3 | 3.71 | 0.28 |

present within the measurement volume during each frame capture. Droplets detected inside the depth of field (DOF) are then measured based on their shadow images, and statistics regarding concentration or size distribution are built by processing a series of images. Worth to mention, there is also an option to measure droplet velocity in the imaging plane by comparing

pairs of consecutive frames and measuring droplet displacement between them. The captured images are either processed in real-time to determine particle positions and sizes (live mode) or stored as graphic files (capture mode). In case of the former, the output is only a list of particles together with their properties (droplet file). Specific quantities included in the droplet file are explained in Table 2. The second complementary output file contains system settings used and measurement summary with some basic statistics of the recorded droplet set (summary file). For the capture mode, one can analyse captured images later

tuning some parameters or access the raw genuine view of the particles (see Fig. 3 in sec. 3.1). However, then the total time of uninterrupted measurement is limited by available computer RAM.

## 2.2 Principle of droplet detection and sizing

The measurement principle applied in VisiSize system was explained by Kashdan et al. (2003, 2004). It stems from the basic observation that with increasing defocus (particle displacement out of the focal plane of the camera lens) the image of the

object is more and more blurred which hinders the proper estimation of the size of the shadow. On the other hand, the range of axial positions guaranteeing acceptable sharpness of the shadow is usually quite limited. As a consequence, the probability of detecting particle inside such restricted SV might be often insufficient to collect meaningful statistics of the suspension in





**Table 2.** Explanation of the parameters reported in output droplet file together with corresponding symbols used to denote them in the text.

| Parameter | Symbol | Units | Description |
|---|---|---|---|
| Frame | $j$ | - | Number of the image containing current particle |
| Particle ID | $i$ | - | Particle number, unique across dataset |
| Area | $A$ | $\mu m^2$ | Estimated cross-sectional area, calculated from $D$ |
| Diameter | $D$ | $\mu m$ | Estimated particle diameter, calculated based on $A_p$ and $A_h$ |
| Shape Factor | $S_f$ | - | Measure of sphericity equal to the ratio of the principle moments of inertia of the shadow, taking values from 0 to 1, with 1 representing a perfect circle |
| X | $x$ | pix | Horizontal position in camera plane |
| Y | $y$ | pix | Vertical position in camera plane |
| Pixel Area | $A_p$ | $pix^2$ | Number of pixels in particle shadow with brightness below threshold $T_p$ |
| Pixel Halo | $A_h$ | $pix^2$ | Number of pixels in particle shadow with brightness between thresholds $T_h$ and $T_p$ where $T_h < T_p$ |
| Distance to Focal Plane | $z$ | $\mu m$ | Estimated axial position in the sample volume, calculated based on $A_p$ and $A_h$; insensitive to direction, thus always positive |

reasonable time. To overcome difficulties described above, Kashdan et al. (2003) applied the method compensating for the effects of imperfect focus. Basically, the displacement from the focal plane is estimated from the degree of image blurring, specifically the gradient of brightness at the edge of inner dark shadow. In their implementation, two threshold limits are determined for each analysed picture based on the histogram of pixel brightness. Both of them lie between the background intensity and the level corresponding to dark centers of particle shadows (see Fig. 2). The upper threshold $T_p$ separates the background from the particle image. All pixels below this value are assumed to belong to effective total particle image area $A_p$ (with the equivalent diameter $D_p$). The lower threshold $T_h$ distinguishes between the dark shadow interior and an outer gray "halo". Pixel with intensities between $T_h$ and $T_p$ are counted to produce the estimate of particle halo area $A_h$ (with the equivalent width $D_h$). Therefore, $A_h$ belongs to $A_p$ and has to be smaller.

With increasing defocus distance, the total area of particle image grows due to blurring until at some point it fades away into background making the object no longer distinguishable. Worth to mention, halo area $A_h$ grows faster than total area $A_p$ because it extends both into outer and inner direction, taking over respectively the background and the interior pixels. Ideally, for object standing exactly in the focal plane $A_h$ should tend to zero. However, it is never the case due to intrinsic diffraction caused by finite aperture of the lens. This effect might be particularly important for small object of the size close to the optical resolution of the system.

Particle total area $A_p$ and halo area $A_h$ can be regarded as directly measured quantities. Both true particle diameter $D$ and estimated defocus distance $z$ are derived from them. The exact conversion is determined with experimental calibration, separately for each lens setting. The pictures of calibration targets of known sizes are taken at known distances away from



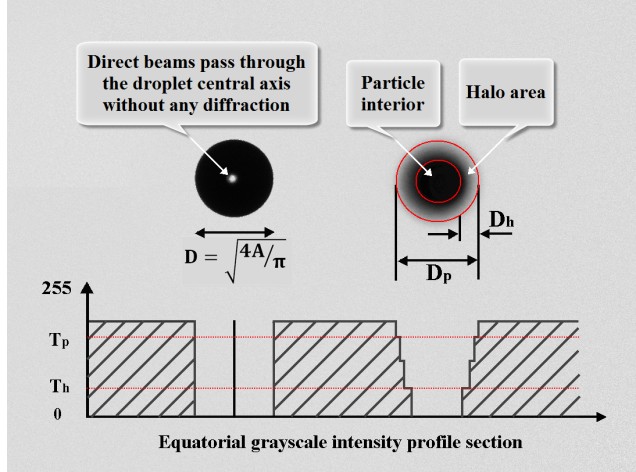

**Figure 2.** Schematic representation of threshold method showing shadow image of two sample droplets, in-focus droplet (left) and defocused droplet (right), alongside grayscale intensity profiles below them illustrating applied thresholds.

the focal plane. Applying the sizing algorithm to the collected data, some function is fitted to approximate the relationship $A_p(D,z)$ and $A_h(D,z)$. For the present VisiSize D30 model, calibration was performed by Oxford Lasers for three possible lens magnification settings (x1, x2 and x4) and incorporated into the software.

In principle, for perfect focusing the halo area $A_h$ should tend to zero, since then the image consists only of a dark disk with sharp edge around. Certainly $A_h$ grows with the distance between actual particle position and the focal plane. The halo is formed exactly around the dark interior of the shadow, i.e. each point on an initially very sharp edge spreads equally in all directions, smoothing the intensity gradient and creating halo of area which scales with the perimeter length and the amount of defocus. For objects with shape close to a circle, the perimeter is proportional to the diameter which leads to a scaling $A_h \sim Dz$ . However, even a point source, while being out of the focal plane is mapped to a circle of confusion in image plane

(camera chip). Therefore $A_h$ should be growing function of z even in case of size $D$ tending to zero. The simple linear relation satisfying the above properties is:

$$A_h = a_1 D z + a_2 z \qquad (1)$$

where $a_1$ and $a_2$ are calibration constants for a given lens magnification.

As described above, with increasing defocus distance $z$, halo around the dark interior develops at the cost of both the

background pixels and the inner shadow pixels. The whole $A_h$ is included in total particle pixel area $A_p$, so only the outward growth affects the value of $A_p$. Outward part of the shell is larger than inward one, hence the proportionality factor would be close to but slightly higher than 0.5. Obviously, in focal plane the image area is dependent on the true cross-section of the physical object, but $A_p$ is composed of pixels of finite size which disturbs the perfect representation of smooth shapes. Consequently, effective diameter length might be diminished by the amount of the order of pixel size. Moreover, even for the

focal plane diffraction comes into play and hinders perfect imaging - objects always appear slightly larger than in reality which





is important especially for small droplets. The enlargement of diameter due to diffraction should be of the order of the optical resolution of the lens system. All the effects can be taken into account in the formula of the form:

$$A_p = a_3 A_h + \frac{\pi}{4} \frac{(D + a_4 + a_5)^2}{\mathrm{pix}^2} \tag{2}$$

where the constants $a_3$, $a_4$, $a_5$ and $\mathrm{pix}$ describe halo blurring, pixelization, diffraction and effective pixel size, respectively.
Note that $A_p$ is specified as the number of pixels, hence the unit conversion factor $\mathrm{pix}^2$ is needed. Having the values of $A_h$ and $A_p$ obtained from the image of the specific particle, both the diameter $D$ and the defocus distance $z$ can be calculated by inverting the Eq. (1) and (2).

## 3   Detection properties

### 3.1   Diagnostic experiment

First diagnostic laboratory experiment was carried out to characterize instrument performance in terms of detection probability and homogeneity which affect statistics of DNC and DSD. A dense stream of poly-disperse water droplets (known to be mostly in the range 2-20 μm in diameter, Korczyk et al. (2012)) was generated with the use of an ultrasonic humidifier. The cloud of droplets was delivered from the humidifier into the SV with 4 cm wide, 70 cm long circular plastic pipe (see Fig. 1). Care was exercised to fill the whole shadowgraph SV with the stream of droplets though both the flow and the particle field were
not exactly homogeneous. The flow velocity was estimated to be of the order of 10 cm s$^{-1}$ and the direction of the flow was aligned horizontally from left to right. For each of the three lens settings (x1, x2, x4) laser power was adjusted to reach optimal background brightness of the pictures and 10 min long measurement in live mode was performed. Figure 3 presents an example image captured during the experiment. Statistics reported by the software for each run are listed in Table 3.

**Table 3.** Statistics of the laboratory experiment with poly-disperse droplets for different lens magnification settings.

| Lens setting | x1 | x2 | x4 |
|---|---|---|---|
| Video frames | 17705 | 17670 | 17695 |
| Empty frames | 3390 | 4508 | 8156 |
| Total counts | 602232 | 951642 | 324625 |
| Counts/frame | 34.0 | 53.9 | 18.3 |
| Min. diameter [μm] | 7.9 | 4.6 | 3.2 |

### 3.2   Focus rejection and depth of field

Results of the experiment show that both halo $A_h$ and total particle area $A_p$ cannot take any values but their admissible range is limited by certain conditions (see Fig. 4). At sufficiently large distance away from the focal plane particle image starts to





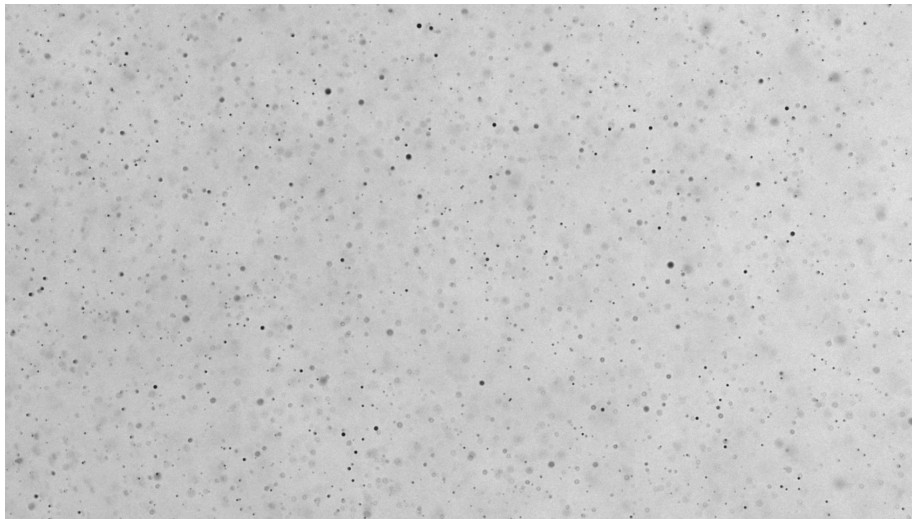

**Figure 3.** A typical shadow image of droplets produced by ultrasonic droplet generator taken during laboratory tests with the camera lens magnification x2.

fade away into background making it no longer distinguishable. Simultaneously, halo takes over almost whole particle image. To avoid measuring objects whose signal-to-noise ratio does not allow for proper sizing, a simple rejection limit is exercised by the software:

$$A_h < 0.95 A_p \tag{3}$$

Importantly, deactivation of this option is not available from the level of the user in the software version 6.5.39.

The condition poses an upper limit for the range of values $A_h$ can take. On the other side, the minimum halo area can be estimated by the diffraction on particle edge because diffraction effects cannot be avoided even in case of the perfect in-focus placement. Indeed, the product of the perimeter length $\pi D$ times diffraction constant $a_5$ approximates the lower limit for halo well (see Fig. 4). Note the large values of $A_h$ are attained only by relatively large droplets and at far defocus. Accordingly, for a given diameter the range of total particle area $A_p$ is also limited. It can grow with defocus distance $z$, but only to the point where halo constitutes 95 % of the image. Otherwise, such particle would have been rejected.

According to Eq. (1) and (2) halo grows linearly with droplet diameter while total image area grows quadratically. It means that for a given defocus distance $z$, halo should constitute larger fraction of the whole image for small droplets in comparison to the large ones. With increasing z the halo would fill the image much sooner in case of small objects and their shadows would sooner fade away into background, making them no longer detectable. Such qualitative reasoning explains the intuitive fact that the range of distance z within which the object can be detected depends on the object size. For instance, effective SV depends on cloud droplet diameter with all the adequate consequences for measuring DSD.

Kashdan et al. (2003) showed that, to a reasonable accuracy, the range of possible defocus distance (depth of field) $[-z_{def}, +z_{def}]$, grows linearly with particle diameter. The proportionality factor comes from the experimental calibration. The measurement





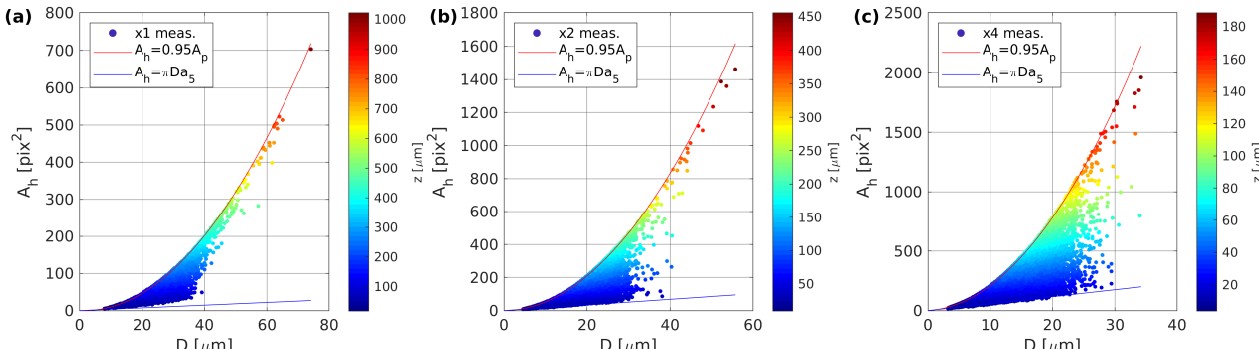

**Figure 4.** Range of halo areas $A_h$ observed in measurement results with changes in diameter size $D$ and defocus distance $z$ for different lens magnification setting: (a) x1, (b) x2, (c) x4.

volume $V$ is then equal to the product of the default DOF $z_{def}$ and effective area of camera sensor $S$ (field of view, FOV).

$$z_{def} = a_6 D \qquad V = 2a_6 DS \tag{4}$$

In a summary file generated by the software, this formula is applied to calculate the volumes $V_k$ corresponding to the consecutive size bins $[D_k, D_{k+1})$, where integers $k = 1, 2, ..., K$ denote bin number.

However, the volumes calculated with such a method are not correct for small particles as typical droplets in atmospheric clouds. The reason for this fact is that the focus rejection condition ($A_h < 0.95A_p$) impose a limitation on the acceptable defocus distance $z$. Hereafter this limit will be denoted $z_{95}$, in contrast to previously introduced $z_{def}$. Specific value can be obtained by using Eq. (1) and (2) to expand inequality of the focus rejection criterion in Eq. (3).

$$z < z_{95} = \frac{0.95}{(1 - 0.95a_3)(a_1 D + a_2)} \frac{\pi}{4} \frac{(D + a_4 + a_5)^2}{\text{pix}^2} \tag{5}$$

The effect of the above condition is illustrated in Fig. 5. Indeed, for all the droplets measured in the diagnostic experiment, Eq. (3) leads to much stronger limitation of the effective SV than Eq. (4). Predicted ranges of defocus distance agree very well with maximum values $z_{max}$ found in the experimental dataset. It can be shown, that focus rejection defines the true SV for particles of small diameter whereas it usually has no effect in the case of large particles like rain drops (then $z_{95} > z_{def}$). The exact critical size depends on the lens settings and the respective calibration. When one considers the relative difference between the two values it can be calculated that it drops below 10 % for particles larger than 210, 260 and 50 μm for lens settings x1, x2 and x4, respectively. Considering the population of such droplets, the choice of the DOF limit would have only minor effect on the results.

### 3.3 Effective sample volumes

Size-dependent DOF has to be accounted for while calculating DSD based on shadowgraph images. Moreover, image processing procedure includes border rejection mechanism which excludes all the objects touching the outer edge of the picture





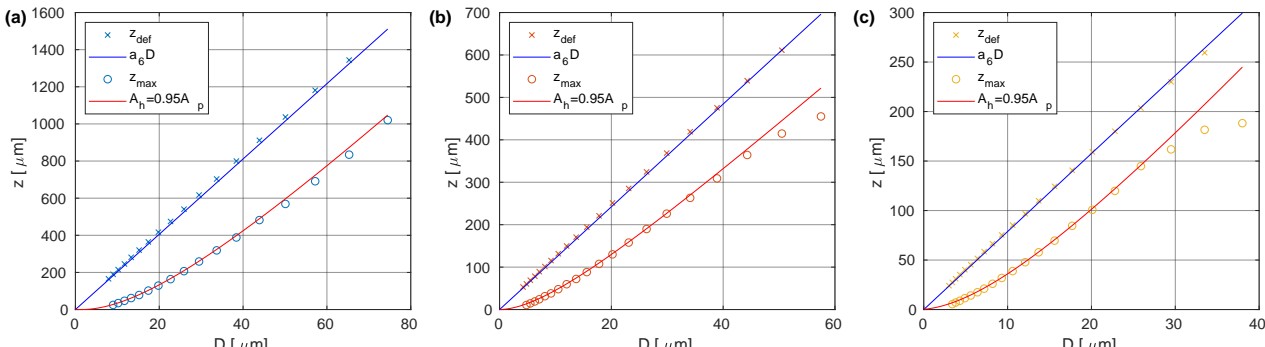

**Figure 5.** Maximum range of defocus distance $z$ for each size bin: derived from the volumes listed in summary file (crosses), and the actual maximum values $z_{max}$ found in the experimental data (circles), together with two analytical approximations - default $z_{def}$ and corrected $z_{95}$. Measurement series for lens magnification settings: (a) x1, (b) x2, (c) x4.

because sizing of such objects would be strongly biased. Thus, the effective cross-sectional area of camera sensor, FOV, is reduced by the margin of the width equal to a diameter under consideration. Cloud droplets are orders of magnitude smaller than the whole FOV, therefore such correction, although reasonable, would not exert much influence on the final results and often might be neglected. Yet, it is significant for large drops, i.e. rain. Eventually, the *default* SVs in calculations of DSD are:

$$V_k^{def} = 2z_{def}\,|_{D_k}\,(L_x - 2D_k)(L_y - 2D_k) \tag{6}$$

where $L_x$ and $L_y$ denote the size of the FOV in $\mu$m and $k$ is the number of a size bin.

However, as explained above the default solution does not account for DOF limitation due to the focus rejection condition. Hence, the *corrected* SVs can be defined as follows.

$$V_k^{cor} = 2z_{95}\,|_{D_k}\,(L_x - 2D_k)(L_y - 2D_k) \tag{7}$$

If the total number of bins $K$ is small or the spread of droplet diameters present in a dataset is particularly large, then the range of sizes for objects belonging to one bin might be significant. For that reason, it would be sensible to introduce SV prescribed exactly for the particular droplet of interest. The method of calculating DSD with the use of *individual* volumes should be more precise from the physical point of view. On the other hand, it requires the access to the list of all droplets whereas corrected and default methods need only the accumulated number of particles within the bins.

$$V_i^{ind} = 2z_{95}\,|_{D_i}\,(L_x - 2D_i)(L_y - 2D_i) \tag{8}$$

According to the above discussion, the effect of choosing different maximum range of defocus distance on effective SV around a droplet is shown schematically in Fig. 6.





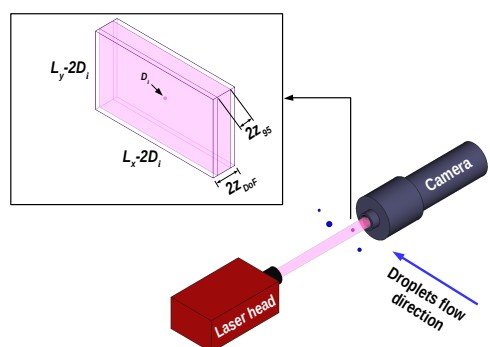

**Figure 6.** Schematic illustrating the difference between the default and the corrected/individual sample volume having the same field of view, $S = (L_x - 2D_i)(L_y - 2D_i)$, but differing in depth of field ($z_{def}$ vs. $z_{95}$).

## 3.4 Correction to concentration and size distribution

Proper quantitative measures of particle concentration and distribution of their sizes in a given suspension should allow for
meaningful comparisons between the measurement series, different instruments and experimental conditions. As explained, effective SVs depend on object sizes. Additionally, practical choice of size bins usually involves widths growing with increasing diameter. In order to characterize droplet spatial arrangement and size differentiation, we introduce a number of counts normalized with respect to both spatial position and size, denoting it as $N_{VD}(x, y, z, D)$. This quantity has units of $\mathrm{mm^{-3}\,\mu m^{-1}}$ and should be interpreted as local probability density function (PDF) of droplet diameter at point $(x, y, z)$ normalized to sum
up to local total droplet concentration (number in a unit volume, DNC) at this point. Then

$$N_{VD}(D) = \frac{\int \int \int N_{VD}(x, y, z, D)dxdydz}{\int \int \int dxdydz} \tag{9}$$

corresponds to the global PDF of droplet diameters and

$$N_V = \int N_{VD}(D)dD \tag{10}$$

to global DNC.

Ideally, $N_{VD}(x, y, z, D)$ should describe physical reality and it cannot be easily estimated from the measurement just by binning the results $(x_i, y_i, z_i, D_i)$, because due to size-dependent DOF the instrument is able to detect the particle of given size $D$ only inside the limited range of $z$. If one divides the ranges of the four variables into bins to construct 4-dimensional grid cells, counts entries in each cell and normalizes by cell 4D volumes, then some of the grid cells in such experimentally obtained $N_{VD}^m(x, y, z, D)$ would be intrinsically missing information. For the same reason, the limits of the integral in Eq. (9)
along $dz$ should depend on $D$.





When DSD is concerned, $N_{VD}(D)$ has an advantage over simple PDF since its values can be compared between measurement series. For the same phenomenon, different lens setting should bring the same results. In order to estimate $N_{VD}(D)$, the number of particles within given size range needs to be divided by respective size-dependent volume. Specifically, for the three methods depicted in the previous subsection:

$$N_{VD}^{def}(k) = \frac{M_k}{F \Delta D_k V_k^{def}} \tag{11}$$

$$N_{VD}^{cor}(k) = \frac{M_k}{F \Delta D_k V_k^{cor}} \tag{12}$$

$$N_{VD}^{ind}(k) = \frac{1}{F \Delta D_k} \sum_{i:D_k \leq D_i < D_{k+1}} \frac{1}{V_i^{ind}} \tag{13}$$

where $M_k = |\{i : D_k \leq D_i < D_{k+1}\}|$ is the number of counts within bin $k$, $\Delta D_k = D_{k+1} - D_k$ is the bin width and $F$ simply the number of frames included in an analysed series.

DSDs obtained with the formulas Eq. (11)-(13) for the laboratory experiment indicate (see Fig. 7) that the values of $N_{VD}(D)$ are significantly underestimated by the default method. It is not surprising, since using Eq. (4) generates SVs which are much larger than the true ones. On the other hand, there is no significant difference between corrected and individual SVs (not shown in the figure). This observation suggests that even without having data for individual droplets usually contained in a particle file, still reasonable DSD can be obtained by correcting accumulated counts listed in a summary file.

Figure 7 compares size distributions of droplets generated with the same device and measured with different lens settings. If each configuration had resolved the whole range of diameters present in a spray, the lines would follow each other. Instead, results approximately agree only for larger droplets which is explained by the instrument resolution improving with the magnification used. The plot suggest that the true minimum droplet size for uniform detection is ∼6 μm for x2 and ∼12 μm for x1. As expected, those values are greater than 3.7 μm and 6.9 μm, respectively, reported by the producer as the resolution

corresponding to the vicinity of the focal plane. Unfortunately, the results do not allow to determine such limit for the lens magnification x4. It can be speculated to equal roughly 4 μm, while definitely stays inside the range defined by the focal plane resolution (about 2 μm) and the above limit for lower magnification (6 μm).

Total DNC $N_V$ can be calculated by suitable integration with respect to the whole range of diameters which in practice turns out into the sum over bins in case of default and corrected method or the sum over individual counts in case of individual

method:

$$N_V = \sum_k N_{VD}(k) \Delta D_k \qquad N_V = \frac{1}{F} \sum_i \frac{1}{V_i^{ind}} \tag{14}$$

Having the information about DNC and DSD, one can calculate simple statistics characterizing the cloud, such as mean droplet diameter $\bar{D}$ or higher order moments of the distribution (mean surface diameter $D_2$, mean volume diameter $D_3$, effective diameter $D_e$). Liquid water content (LWC) can be estimated by summing up volumes of the droplets measured.

Because the DSD differs between the methods and lens settings, the resulting mean diameters and LWC will also vary. Table 4 summarizes the results obtained for the poly-disperse stream produced by an ultrasonic generator. DNC is about three to four





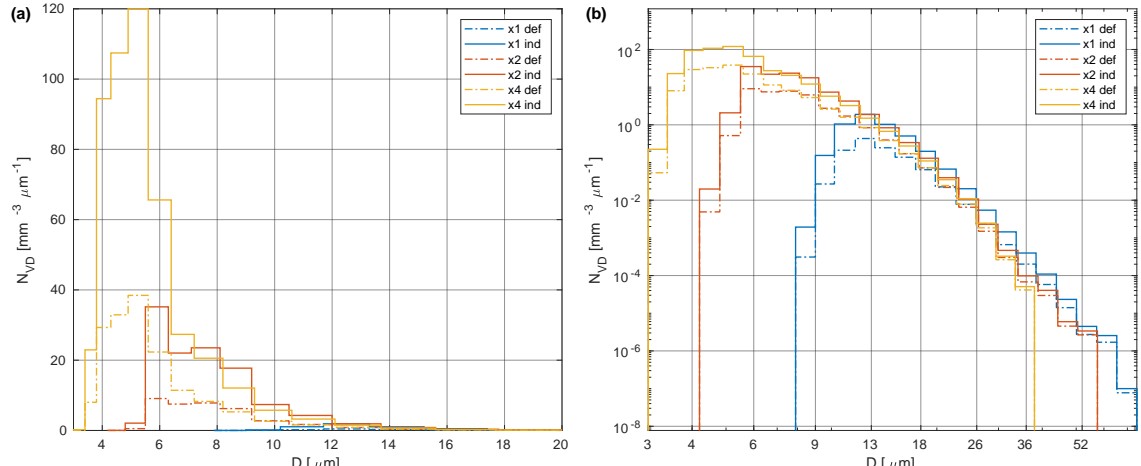

**Figure 7.** Comparison of droplet size distributions obtained with two methods, default (def) and individual (ind), in the course of the three measurement series differing in lens magnification settings. Same data plotted in (a) linear and (b) logarithmic scale.

times larger for corrected and individual method with respect to the default one. In all the cases, statistics of diameter are smaller for new methods in comparison to the default. It can be explained by noting that the largest relative difference in SVs between the methods appears for smallest droplets (see Fig. 5), so it is their contribution to the DSD which changes at most.

**Table 4.** Results of the laboratory experiment with poly-disperse droplets: comparison of different methods used for estimating size-dependent sample volumes.

| Lens setting | x1 | | | x2 | | | x4 | | |
|---|---|---|---|---|---|---|---|---|---|
| **Method** | def | cor | ind | def | cor | ind | def | cor | ind |
| $N_V$ [mm$^{-3}$] | 2.1 | 8.6 | 8.6 | 37.3 | 112.2 | 112.3 | 114.1 | 331.6 | 330.1 |
| $\bar{D}$ [µm] | 14.1 | 13.7 | 13.7 | 8.3 | 7.9 | 7.9 | 6.0 | 5.7 | 5.7 |
| $D_2$ [µm] | 14.5 | 13.9 | 14.0 | 8.6 | 8.2 | 8.3 | 6.4 | 6.0 | 6.0 |
| $D_3$ [µm] | 14.9 | 14.2 | 14.3 | 9.1 | 8.6 | 8.6 | 6.9 | 6.4 | 6.4 |
| $D_e = D_3^3/D_2^2$ [µm] | 15.7 | 14.9 | 15.0 | 10.0 | 9.4 | 9.4 | 8.0 | 7.3 | 7.3 |
| LWC [g m$^{-3}$] | 3.6 | 13.0 | 13.2 | 14.5 | 37.3 | 37.7 | 19.6 | 45.7 | 46.2 |

**3.5 Homogeneity of detection**

One of the important properties characterizing particle sizing instrument, apart from the extent of the SV, is homogeneity of detection inside that volume. Perfect homogeneity can be defined as the case when the probability of detecting the given





particle of interest is the same everywhere, as long as the particle appears to stand at the position belonging to the respective SV, regardless of how complicated its boundaries are. If the probability depends on the position in space, the results obtained

with the system might be biased. It is particularly important for calculating the measures of droplet spatial arrangement, e.g. nearest neighbour distance or radial distribution function (Larsen and Shaw, 2018).

Within the present study, quality of detection is evaluated for shadowgraph system VisiSize D30 based on the long record of the plume of water droplet produced by an ultrasonic droplet generator. As stated earlier (sec. 3.1), the stream of droplets was supposed to extend further than the SV in each direction. Unfortunately, no independent information is available about

whether the DNC inside the visible plume was uniform. One can expect that it drops from the maximal value in the center towards the edges where the intensive mixing takes place between cloudy and clear air portions. Nevertheless, the mixing zone was observed to be outside the SV - at least on average, because in general the outflow was quite dynamic. Therefore, statistics integrated over time will be analysed in this section. To allow for drawing conclusions regarding the detection, it is assumed that on average the real physical conditions are homogeneous, i.e. during the experiment the flow filled each part of the entire

SV with identical concentration of droplets having identical properties (e.g. DSD).

Indeed, results of the experiment integrated over size, time and distance to the focal plane mostly provide relatively constant average droplet concentration with respect to the two principal directions of the camera sensor - horizontal and vertical. The calculated average concentration falls down significantly close to the edges of the FOV (not shown) which is expected concerning border rejection procedure. Figure 8 in panels (a) and (b) shows droplet concentration $N_V(x)$ and $N_V(y)$ suitably

integrated over size and over other dimensions. The values mostly decrease gradually from the center to the sides in case of horizontal direction and from the bottom to the top in case of vertical. The relative differences are up to about 10 %, i.e. small enough to be as well assigned to the non-uniformities of the plume instead of instrumental flaws. However, in case of the series recorded with lens setting x1, the concentration falls with height $y$ by a factor of more than 10 from the bottom to the top of the FOV. We speculate this effect is of instrumental origin, as the difference is rather too large (almost exponential) and the

timescale too short to allow for the explanation by possible gravity sorting of the droplets in the plume. Probable reason is nonuniform illumination of the scene. Although non-uniformities can be compensated for by background normalization (such function is built-in in the software), signal-to-noise ratio of an individual particle shadow might still depend on the position with consequences for detection probability. Due to extensive halo some of the particles might have been rejected. Such problem with illumination might stem from imperfect manual alignment between the laser and the camera, yet we note achieving

the satisfactory light conditions with lens setting x1 and lower is challenging. There are no such issues for higher settings, as the FOV is smaller then and it fits easily inside the uniform core of the laser beam.

Axial dependence is more difficult to evaluate, because the extent of the acceptable DOF depends on both particle size and lens setting. As expected, average concentration decreases with $z$, since for further distances only the droplets which are large enough can be counted (see Fig. 8 panel (c)). This trend is a result of coexisting effects of DOF limitation and the shape of

real DSD where large droplets constitute minor part of the total number. Interestingly, the first $z$ range contains much smaller number of droplets than the second, regardless of the lens setting. Such unphysical behaviour might stem from the sizing algorithm, which calculates $z$ position out of the halo area. Small particles, like the ones observed in the experiment, are





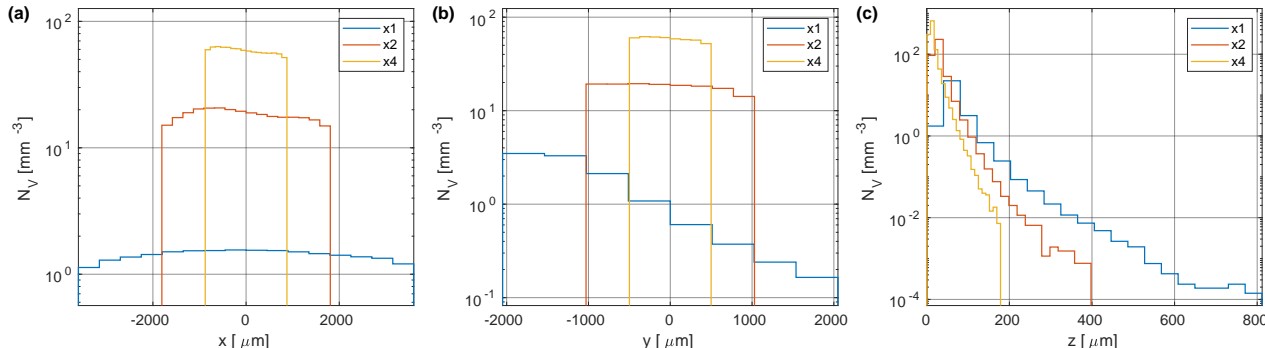

**Figure 8.** Dependence of total droplet concentration on (a) horizontal, (b) vertical and (c) axial position.

blurred even in the focal plane which makes them being faulty positioned at higher distances than the true ones. Equation (1) assumes no halo for the focal plane $z = 0$ which cannot be realized due to diffraction. Then, non-zero halo $A_h$ is measured and

it results in $z > 0$ for all the shadows.

Moreover, it can be noted that for larger distances $z$ the concentration rises with pixel size (falls with magnification, dependent on the lens setting chosen, see Table 1). This fact refers to the DOF limits (Eq. (4)) and respective curves plotted in Fig. 5. Comparison between the three magnifications implies that for any fixed distance $z$ higher than roughly $117\,\mu m$, the lower limit of the detectable size range increases with the magnification, i.e. wider spectrum of droplets can be counted at setting x1 than

x2 and x4. For example, at $z = 150\,\mu m$, the lens setting x1, x2 and x4 allow for the detection of droplets larger than 21.4, 22.2, $26.4\,\mu m$ in diameter, respectively.

However, the probability of detection might depend both on the position in space and on the particle size. Therefore, cross-correlated dependence was examined and illustrated in the form of 2-dimensional $(z, d)$ maps in Fig. 9. It is easy to recognize the limitation introduced by the focus rejection criterion which also controls the effective SV (red line). What is more, the

DSD seems to change with the distance $z$ from the focal plane, in particular for the lens setting x1. Precisely, it is a simple consequence of the focus rejection criterion analysed above that the increasing portion of smaller sizes are not detected with increasing distance $z$ but the concentration of sizes which are well above this limit should ideally not change with the distance along the optical axis of the system. It clearly does in case of lens setting x1. Large droplets are present only far from the focal plane and the concentration of droplets of a given size grows with $z$. Such behaviour suggests that the sizing procedure for the

measurement with lens setting x1 might have not work properly. Possibly, due to insufficient brightness of the pictures the halo area was overestimated at the cost of inner particle shadow. According to sizing Eq. (1) and (2), this leads to overestimation of defocus distance together with underestimation of the diameter.

As a consequence, we discourage using lens setting x1 in further studies of cloud droplets. The large minimum particle size for uniform detection reported earlier ($\sim 12\,\mu m$) makes this option of limited utility anyway. Fortunately, in experimental runs

with higher magnifications, i.e. lens setting x2 and x4, the DSD behaves like expected – large droplets appear with similar concentration both in the middle and at the sidelines of the SV. The occurrence of the smaller ones is limited by the focus





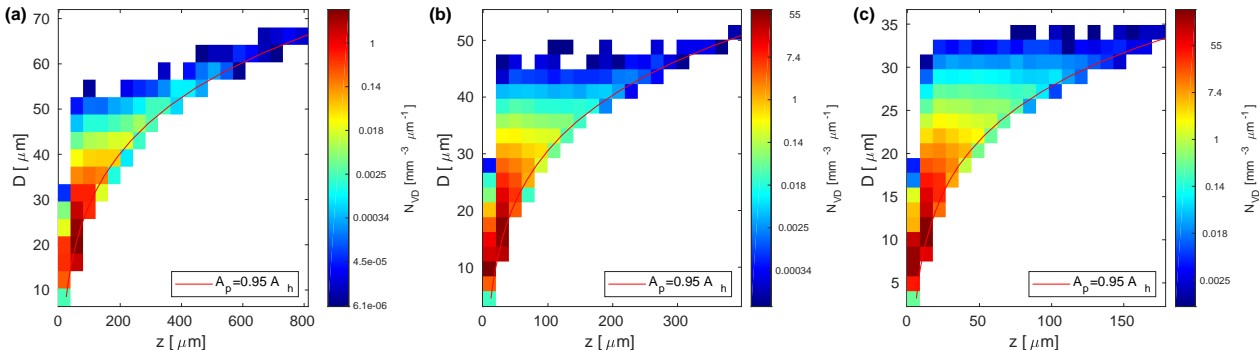

**Figure 9.** Variability of droplet detection properties in size-space domain for lens setting (a) x1, (b) x2, (c) x4.

rejection criterion (red line). There is only a discrepancy between first and second $z$-bin. The same effect was noticed in Fig. 8 panel (c).

## 4   Sizing accuracy

### 4.1   Diagnostic experiment

Second diagnostic experiment was carried out to characterize instrument performance in terms of particle sizing which affects the relevant statistics of cloud droplets including mean droplet diameter and DSD. Monodisperse water droplets were generated with Flow Focusing Monodisperse Aerosol Generator 1520 (FMAG) manufactured by TSI Inc. and measured with VisiSize D30. FMAG uses periodic mechanical vibration to break a narrow jet of a liquid into droplets of desired size (within the range of 15 to 72 μm in diameter). Efficiency and accuracy of droplet generation is enhanced by aerodynamic flow focusing and charge neutralizer making FMAG a common tool applied for calibration of aerosol spectrometers and droplet sizing instruments (Duan et al., 2016).

In the course of the experiment, pressurized $N_2$ (1.0 psi) was used as the flow focusing gas, and Ultra-purified $H_2O$ as the liquid. Droplet size was controlled by adjusting two parameters: liquid flow rate and vibration frequency. Three different settings were applied resulting in droplet diameters of 20.13, 39.25 and 57.55 μm as obtained with the formula provided by the manufacturer (see the settings listed in Table 5). Geometrical standard deviation among the generated droplet population is supposed to be 1.05 or smaller (Duan et al., 2016).

Experimental setup was arranged in two different configurations (see Fig. 10). The first without a cylindrical case over the nozzle, with shadowgraph SV as close as possible to the nozzle head (about 4.5 cm). The second with the case and the dilution air (flow rate of 5 L min$^{-1}$) forcing the droplets to leave the cylinder. The shadowgraph SV was then located above the cylinder exit (about 18.5 cm over the nozzle head).





**Table 5.** Settings of the FMAG droplet generator used during sizing tests.

| FMAG 1520 settings | | | |
|---|---|---|---|
| Inlet air [psi] | 10.0 | | |
| Dilution air [L min$^{-1}$] | 5.0 | | |
| Focusing air [psi] | 1.0 | | |
| Flow rate [mL h$^{-1}$] | 2.0 | 8.0 | 18.0 |
| Vibration frequency [kHz] | 130 | 70 | 50 |
| Droplet size [µm] | 20.13 | 39.25 | 57.55 |

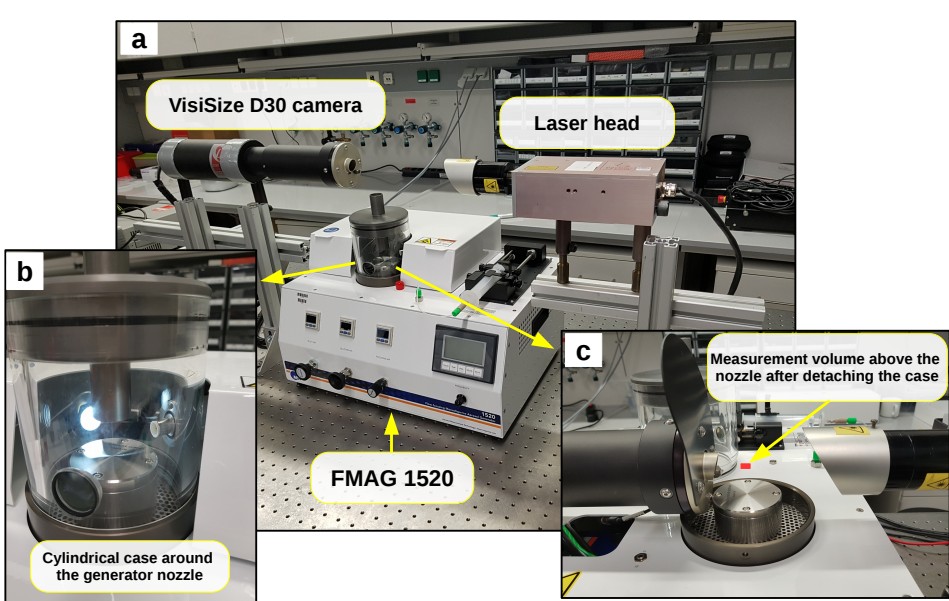

**Figure 10.** Experimental setup for studying sizing performance: VisiSize D30 located above the exit of FMAG in order to measure outgoing droplets (a) in configuration with the cylindrical case above the nozzle (b) or without it (c).

## 4.2  Results

Sample shadow images of droplets produced by the FMAG and captured by the VisiSize D30 are reproduced in Fig. 11. The configuration of the setup was as in panel (c) of Fig. 10, i.e. without the case covering the nozzle. Despite the population of





droplets is supposed to be mono-disperse, one can see the range of sizes. For each droplet size specified by the parameters of
FMAG, a series of images was taken by the shadowgraph with three different lens magnification settings. Results are presented
in Fig. 12 in the form of probability density functions (PDF) (equivalent to $N_{VD}(D)/N_V$). Such measure was chosen in order
to compare the DSD between different magnifications and measurements substantially different in total droplet concentration
which itself is not quantity of interest in the current analysis.

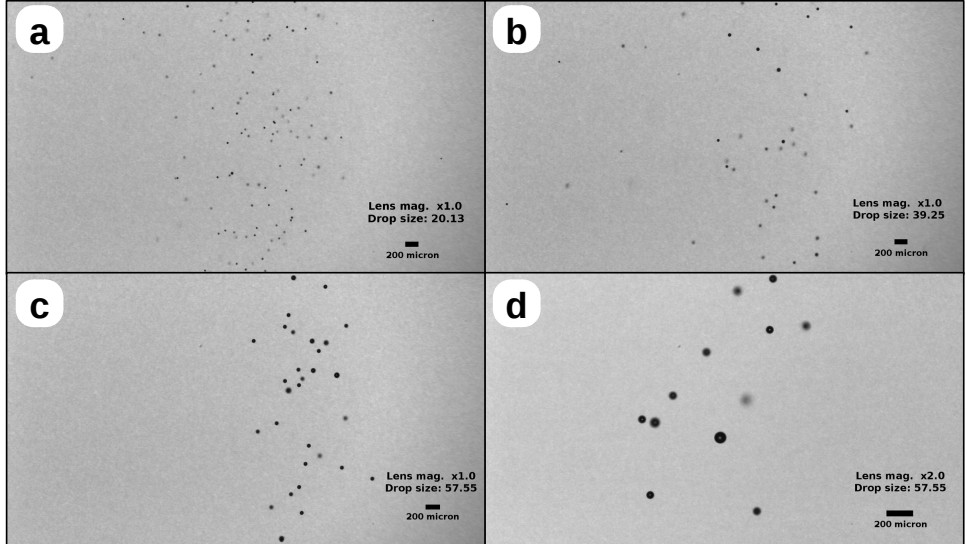

**Figure 11.** Sample shadow images captured by VisiSize D30 showing droplets of different size generated by FMAG.

Strikingly, all the histograms contain multiple peaks which suggests quite frequent collisions between droplets on their way
from the nozzle to the SV of the shadowgraph. The position of the first peak corresponds well with the generated droplet size
in most of the cases. Its width can be attributed to the inevitable spread of true generated droplet diameters as well as to the
imperfect imaging and sizing of droplet shadows. Left-side skewness of the tails implies partial evaporation.

Taking into account geometric standard deviation of generated droplets (1.05), histogram bin width (0.5 μm) and evaporation,
the sizing by shadowgraph is pretty accurate. Only in case of smallest droplets (20.13 μm) and lowest lens magnification (x1)
the reported diameters seem to be significantly biased. This issue is probably of instrumental origin and might correspond
to non-uniform illumination of the FOV which also caused inhomogeneity of detection of relatively small objects for that
particular lens setting (see sec. 3.5).

The fact of droplet collisions is further corroborated with measurements conducted in the configuration as in Fig. 10
panel (b), i.e. with the cylindrical case over the nozzle and longer distance between nozzle exit and the shadowgraph SV
(around 18.5 cm). Longer path enhances the chance of collisions and effects of evaporation. The former can be clearly seen in
histograms presented in Fig. 13. Consecutive peaks correspond to single generated droplet, double collision and triple collision





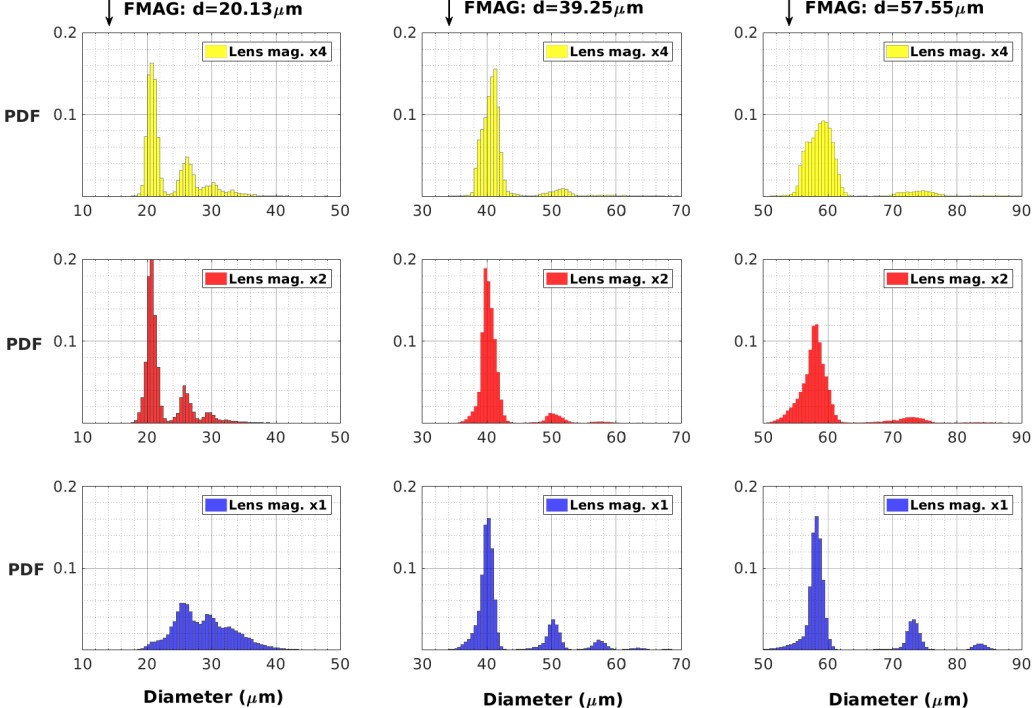

**Figure 12.** Probability distributions of droplet size measured by VisiSize D30 in configuration as in fig. 10 panel (c) (distance between sample volume and nozzle head is $\sim 4.5\,\mathrm{cm}$), for different camera lens magnifications and different FMAG output droplet sizes.

as calculated simply by summing the volumes. Here, lens magnification setting x1 was selected due to largest SV, thus best statistics which can provide better estimation of less frequent events in the probability distribution. More distant peaks can be

traced better in case of largest droplet size suggesting the probability of high-order collisions might increase with size. This can be expected recalling extensive studies on cloud droplets collision kernels (Devenish et al., 2012; Grabowski and Wang, 2013). Left-skewed tails are also visible, in particular for the first peak, which is the consequence of evaporation on the way of $18.5\,\mathrm{cm}$ from the nozzle.

Finally, the accuracy of sizing is evaluated more quantitatively in the scatter plot in Fig. 14 which compares initial droplet

size as specified on FMAG with the results obtained with the shadowgraph – mean droplet diameter reported by the software and the position of the first major peak in the size histogram. Obviously, mean diameter is larger than the first peak as it includes the contribution of collided droplets. Excluding the problematic case of magnification x1 for size $20.13\,\mu\mathrm{m}$, the position of the first peak deviates up to $2\,\mu\mathrm{m}$ from the original value and the relative error ranges up to roughly 5 %. However, with regard to the accuracy of shadowgraph instrument, it is only an estimation of the upper bound for those quantities, as the generated size

is also subject to intrinsic uncertainty. Hence, as far as the comparison with FMAG allows to say, the sizing by shadowgraph is indeed pretty accurate, apart from small droplets observed with magnification x1. Interestingly, in most cases the first peak





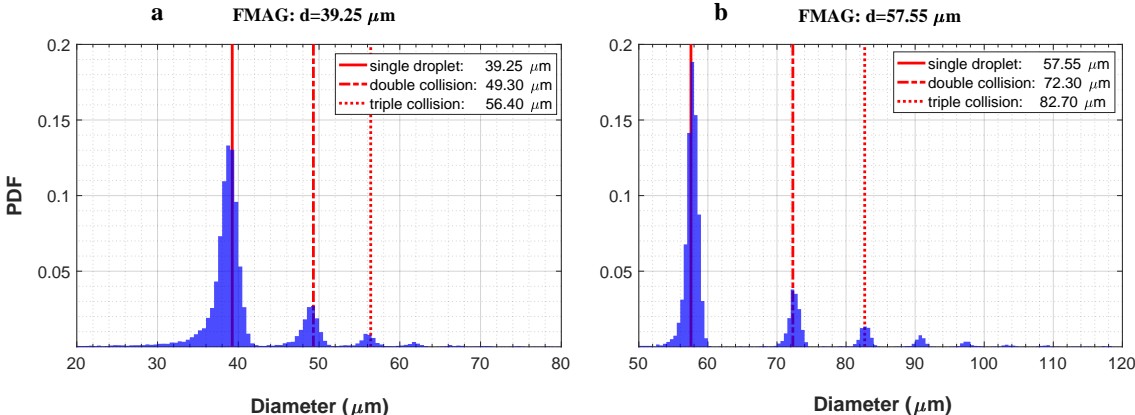

**Figure 13.** Probability distributions of droplet size measured by VisiSize D30 in configuration as in fig. 10 panel (b) (distance between sample volume and nozzle head is ∼18.5 cm), for camera lens magnification setting x1. Initial size of the generated droplets as well as the size after double and triple collisions are marked with vertical red lines.

estimation is higher than the FMAG one while the opposite can be expected due to evaporation effects. This fact suggests that the shadowgraph slightly overestimates the sizes in relation to FMAG in most cases.

## 5  Field measurements

After laboratory tests, the shadowgraph VisiSize D30 has been used for the first time to measure droplets in atmospheric clouds – in order to test the instrument performance under harsh environmental conditions, compare it with other probes already in service in cloud physics studies, and to study microphysical properties of warm (liquid) orographic clouds. The measurements were performed in a ground-based mountain observatory – Environmental Research Station (Umweltforschungsstation) Schneefernerhaus located on the southern slope of Zugspitze in Bavarian Alps – during two observational periods in July

and August 2019. Typical meteorological conditions at this place together with respective cloud and turbulence properties are described in detail by Siebert et al. (2015) and Risius et al. (2015).

Comprehensive analysis of the field experiment alongside with the results obtained with the shadowgraphy imaging technique is going to be covered in a separate article while here we present example observations of cloud microphysical properties representing the range of conditions typical at the place. The data was collected on 13 July 2019 when the clouds covered

the sky for most of the day (7-8 oct). However, due to the wind and complex terrain the observatory is usually exposed only to intermittent portions of cloudy air. Two measurements series, each 15 min long, recorded within relatively homogeneous conditions were selected. The first was performed in the afternoon (14:46-15:01 LT) using lens magnification setting x2, the





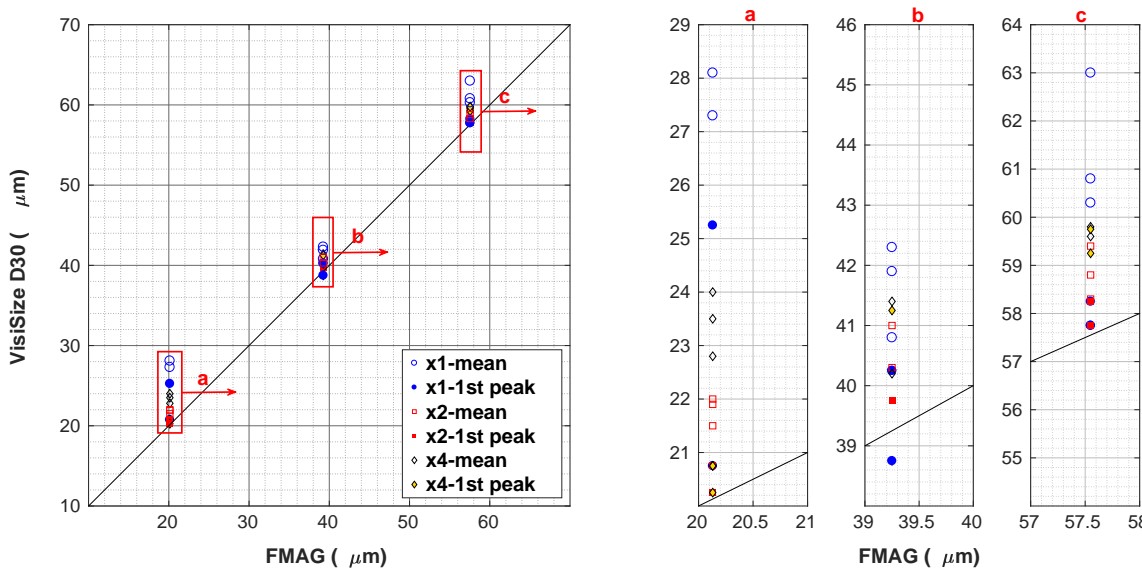

**Figure 14.** Comparison of droplet sizing between VisiSize D30 and FMAG: scatter plot illustrating the position of the first peak in size histogram and arithmetic mean diameter with respect to FMAG generated droplet sizes for different lens magnification settings (left). Red boxes (a, b, c) are enlarged in panels on the right.

second in the late evening (23:19-23:34 LT) using x4. Throughout the day temperature varied around 0° C. Wind was predominantly westerly, coming over the saddle in the mountain range located west from the observatory. It was stronger for the first

measurement series, with velocity of around 5 m s$^{-1}$ and fluctuations of 2 m s$^{-1}$, than for the second one (about 1.5 m s$^{-1}$ and 1 m s$^{-1}$ of mean velocity and fluctuations, respectively). There was no precipitation noticed in the afternoon, while light rain occurred in the evening shortly before the measurement.

Figure 15 presents normalized DSDs calculated with the methods described in sec. 3.4 for the two recorded events. Basic microphysical statistics calculated based on the DSDs are listed in Table 6. Naturally, the results cannot be directly compared

between the series as the conditions were slightly different. However, DNC as well as mean diameter and LWC resemble typical properties of non-precipitating continental clouds and stay close to the range of conditions reported by Siebert et al. (2015) for Schneefernerhaus observatory based on the measurements with completely different instrument (Phase Doppler Interferometer).

The cloud observed in the afternoon seems to contain more significant portion of larger droplets (10-15 μm) with respect to

the evening one. Yet, in both cases the droplets were produced mostly by condensation as the maximum diameters measured do not correspond to the sizes capable of efficient rain formation by collisions. It should be noted, that moments of the DSD (mean diameter statistics) are most probably moderately overestimated because the portion of small droplets might not be properly detected in the whole relevant volume. This limit of minimum diameter for uniform reliable detection was estimated to be ∼6 μm for lens setting x2 (see sec. 3.4) and unfortunately not found exactly for lens setting x4 (though definitely stays





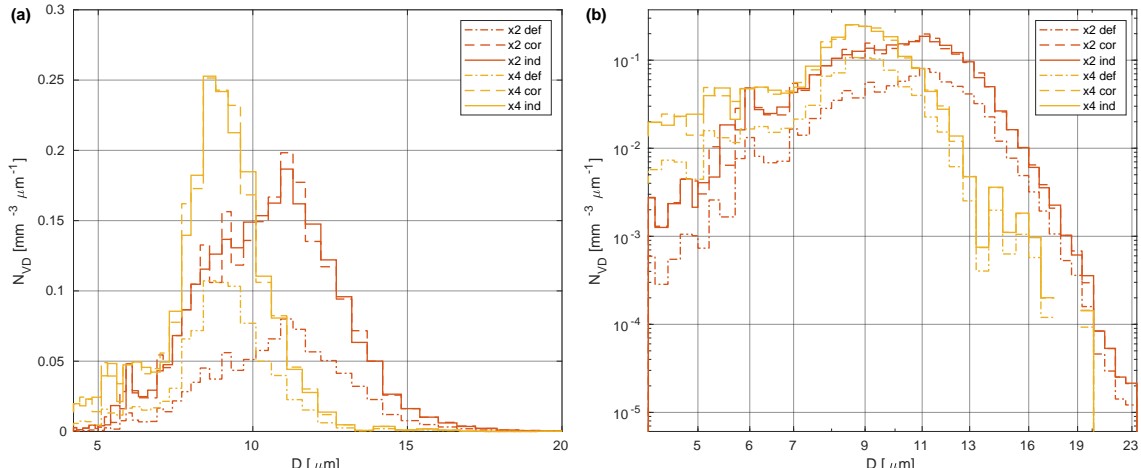

**Figure 15.** Droplet size distributions observed with the VisiSize D30 in clouds at Schneefernerhaus observatory during two events on 13 July 2019 - afternoon (14:46-15:01 LT) and late evening (23:19-23:34 LT), the former taken with lens setting x2, the latter x4. Three methods of calculating the results are marked with different linestyle: default (def), corrected (cor) and individual (ind) (see sec. 3.4 for explanation). Same data plotted in (a) linear and (b) logarithmic scale.

**Table 6.** Results of the cloud observations performed with the VisiSize D30 at Schneefernerhaus on 13 July 2019.

| Time (Lens setting) | 14:46-15:01 LT (mag. x2) | | | 23:19-23:34 LT (mag. x4) | | |
|---|---|---|---|---|---|---|
| **Method** | def | cor | ind | def | cor | ind |
| $N_V$ [cm$^{-3}$] | 359.4 | 927.0 | 928.7 | 341.6 | 797.9 | 797.8 |
| $\bar{D}$ [μm] | 10.8 | 10.6 | 10.6 | 9.0 | 8.8 | 8.8 |
| $D_2$ [μm] | 11.0 | 10.8 | 10.8 | 9.1 | 9.0 | 9.0 |
| $D_3$ [μm] | 11.3 | 11.0 | 11.0 | 9.3 | 9.1 | 9.1 |
| $D_e = D_3^3/D_2^2$ [μm] | 11.7 | 11.5 | 11.4 | 9.6 | 9.4 | 9.5 |
| LWC [g m$^{-3}$] | 0.27 | 0.65 | 0.65 | 0.14 | 0.32 | 0.32 |

between 2 and 6 μm). Obviously, the degree of the bias decreases with magnification (improving resolution). For the same reason, the total DNC might be moderately underestimated with respect to the true one. Nevertheless, all the instruments suffers from similar issues whenever the range of detectable diameters is finite. The relative differences between the three methods of DSD calculation are very similar to what was stated in case of laboratory experiment (sec. 3.4).

     The comparison between the two example observations discussed in this section also illustrates the trade-off regarding the

choice of magnification. Larger one (x4) provides better resolution and proper representation of the left tail of the DSD (below


6 μm), though the right tail of relatively scarce large droplets is then poorly statistically represented because the total number of counts is quite modest. Specifically, despite the similar DNC and duration of the measurement, roughly 10 times more droplets counts were recorded in the first series (x2), simply due to the larger SV.

## 6 Conclusions

420 The shadowgraph imaging system – Oxford Lasers VisiSize D30 – has been tested and characterized with respect to cloud microphysical measurements, i.e. number concentration and size of cloud droplets. The instrument captures images containing shadows of multiple particles, counts them and estimates sizes correcting for image blurring due to out-of-focus position. Although developed for industrial applications, the system can be applied for cloud physics studies. Nevertheless, diagnostic laboratory experiments pointed out important limitations which need to be considered.

425 First, the sample volume within which a droplet is detectable depends on its size because blurring caused by defocus differently affects images of particles of different size. This fact has to be always taken into account when estimating droplet concentration (in a unit volume). The solution implemented in the software assumes linear relation between depth of field and particle diameter which is efficient for relatively large objects (>260 μm, exact value depends on the lens magnification). However, in case of small droplets, like the cloud ones, additional focus rejection criterion impose much stronger limit on 430 acceptable depth of field. It affects relevant sample volume and leads to underestimated number concentration. Therefore, we developed correction method using sample volume based on that limit.

Second, the analysis of detections in a dense poly-disperse stream of droplets implied that the minimum droplet size for reliably uniform detection is significantly larger than the resolution in the focal plane. It was estimated to be ∼6 μm and ∼12 μm for the camera lens magnification settings x2 and x1, respectively. Potentially, it can be enhanced by careful data 435 conditioning, i.e. with strong limit on the estimated distance from the focal plane, but at the cost of decreasing sample volume. Furthermore, detection probability was found to be satisfactorily homogeneous across the field of view, except for the small magnification setting (x1). Minor issues were revealed with respect to axial direction, probably caused by substantial diffraction effects on small droplets.

Third, the test of sizing accuracy was performed using mono-disperse droplet generator (FMAG 1520, TSI). Substantial 440 effects of droplet collisions and evaporation were observed in the size histograms obtained with the shadowgraph. Notwithstanding, after filtering out collisions by selecting first major peak the results indicated reasonable agreement between diameters reported by the shadowgraph and those supposed to be generated by the FMAG. The relative difference was not larger than 2 μm or 5 %, again except for the lens magnification setting x1 which caused difficulty in uniform illumination of the scene.

Finally, the system under study was applied to sample atmospheric clouds in a ground-based mountain observatory. It per- 445 formed satisfactorily well under windy, cloudy, humid weather conditions and provided quite an extensive set of microphysical data which is intended to be presented and discussed in detail in a separate publication. The results of selected observations analysed here comply with the expected conditions and previous independent measurements performed at that location.





To sum up, the VisiSize D30 can be successfully applied for cloud microphysical measurements. However, relevant quantities like droplet size distribution, number concentration, mean diameter or effective diameter need to be calculated with care, accounting for size-dependent sample volume. While conducting the experiment, one should pay an attention to appropriate adjustments of the laser and the camera in order to assure uniform illumination of the field of view. We recommend to avoid low magnifications (e.g. x1) as they make the proper illumination adjustments more difficult and are of limited utility for cloud studies due to large limit on minimum diameter for satisfactory detection. Those might be instead advantageous for sampling drizzle and rain which is a topic currently under study.

*Author contributions.* J.L.N., M.M. and S.P.M. designed the study. J.L.N. planned and carried out the detection experiment (sec. 3). M.M. planned and carried out the sizing experiment (sec. 4). M.M. carried out the field experiment, while M.M. and J.L.N. analysed the collected data. J.L.N. and M.M. wrote the manuscript with contributions from S.P.M.

*Competing interests.* The authors declare that they have no conflict of interest.

*Acknowledgements.* We thank Dr. Gholamhossein Bagheri, Dr. Jan Molacek and the staff of Max-Planck Institute for Dynamics and Self-Organisation (MPIDS) for providing FMAG aerosol generator and the help during the sizig laboratory experiments in Goettingen. We are also grateful to Dr. Till Rehm and the staff of Umweltforschungsstation Schneefernerhaus (UFS) for their help during the field measurements at Mt. Zugspitze. Special appreciation is given to Dr. Wojciech Kumala for the technical support in the course of detection laboratory experiments in Warsaw. The authors would like to acknowledge the financial support for this work, provided by the Marie-Sklodowska Curie Actions (MSCA) under the European Union's Horizon 2020 research and innovation programme (grant agreement n°675675).





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
