# Peer review of "Applicability of the VisiSize D30 shadowgraph system for cloud microphysical measurements"

_Atmospheric Measurement Techniques, 2020_

## Referee Comment (RC1) · Anonymous Referee #2 · 17 Nov 2020

The authors of this study test, characterize and apply a commercial shadowgraph system, which has been originally designed to characterize industrial and agricultural sprays, for measuring cloud microphysical properties, such as droplet size distribution and number concentration in atmospheric clouds. Due to laboratory investigations, among others, the authors improved estimations of droplet number concentration and size distribution as well as verified the sizing accuracy. Then, the system was tested in a real atmospheric cloud. They finally conclude that the shadowgraph system can be applied for cloud microphysical measurements if certain features of the system are considered, such as the droplet size-dependent measuring volume.

[Figure]

The study is very important for the atmospheric cloud community as it presents an additional instrument option for obtaining cloud microphysical properties such as cloud size distribution and its number concentration. I recommend publication after the following comments have been addressed.

General comments:

Would it be possible to use the shadowgraph system to obtain some 3D information about the position of the cloud droplets?

Could you give any estimate for the largest detectable number concentration? In case of large number concentrations, for example, it might be possible that droplets "hide" one after the other.

Concerning the application of the shadowgraph system in the atmospheric cloud (Section 5): Has there been any other cloud probe which measured the size distribution at the same time as the shadowgraph system? If so, could you present a comparison of the obtained droplet size distributions and its moments, respectively? If not, I would suggest, for a future study, to have a comparison to another cloud probe.

Specific comments:

Page 7, line 145-146: Looking on Fig. 1, I would think that the direction of the droplet flow was vertically aligned.

Fig. 5c: For diameters larger than $30\mu$m, z_max deviates significantly from the analytical approximation z_95. Do you have any explanation for that?

Fig. 7 and Fig. 15 and the corresponding text: In Fig. 7 only DSDs applying methods "def" and "ind" are shown, for given reasons. However, later in Fig. 15, DSDs for "def", "cor" and "ind" are shown. Wouldn't it make more sense, to show DSDs for methods "def", "cor" and "ind" in Fig. 7 and explain that there are only small differences between methods "cor" and "ind" and then only show DSDs for "def" and "ind" in Fig. 15?

[Figure]

Page 14, line 280: This gradual decrease from the center to the sides is obvious for x1, but not really for x2 and x4. Could you please comment on that?

Page 14, line 284-291: I think the speculation given here is reasonable. However, could this also be true for x2 and x4? But here more (smaller) droplets are detected, both in a smaller FOV compared to x1 which compensates this feature?

Page 15, Line 318: Could you please provide a size range where you would not use x1. As later said, x1 makes sense for larger droplets in the drizzle and rain size range. Please make clearer under which circumstances you would avoid using x1, and vice versa.

Fig. 9 and respective text: Does the discrepancy in the first and second z-bin has any consequences on the calculation of the DSD and DNC?

Minor comments:

Page 2, Line 50: I would suggest to write either "shadowgraphy" or "the shadowgraph technique".

Page 3, line 70: Please add "the" at the beginning of the sentence: "The two main parts [. . .]".

Page 6, line 120: Since "z" is a parameter it should be given in italic type font.

Page 7, line 137: Please write "[. . .] inverting Eqs. (1) and (2)".

Page 7, line 142: It should read "[. . .] in the range of 2 -20 $\mu$m [. . .]".

Page 7, line 143-144: I would suggest to write: "Care was taken to fill [. . .]".

Page 8, line 169: Please add "DOF" in the brackets behind "(depth of field)".

Fig. 14: I would suggest to increase the size of the symbols in the figures here.

[Figure]

---

## Referee Comment (RC2) · Anonymous Referee #1 · 18 Nov 2020

This paper introduces the commercial shadowgraph system VisiSize D30 and assesses its applicability for microphysical measurements of cloud properties (e.g. droplet number concentration and size distribution). The depth of field and sample volume of the instrument were characterized using a dense stream of poly-disperse particles. Furthermore, laboratory experiments with mono-dispersed droplets were performed for particle sizing. Based on these experiments, the authors developed correction methods to improve estimations of number concentration and size distribution. Finally, the instrument was applied in real atmospheric clouds at a mountain observatory.

This paper highlights the advantages and limitations of the shadowgraph technique for cloud research and proposes methods, which are interesting for cloud probes with a size-dependent sample volume. I recommend publication after addressing the comments below.

**General comments**

It is challenging to characterize the sample volume of the shadowgraph system (Sect. 3) without accurate information regarding the properties of the plume (e.g. uniformity, size distribution of poly-dispersed droplets). Uncertainties in the calibration method can have large uncertainties on the droplet number concentration and size distribution. The assumptions and speculations in Sect. 3 are all justified, but some uncertainties remain. For example, the different lens systems show different dependencies of the number concentration on the horizontal, vertical and axial position (Fig. 8), some of which are attributed to the non-uniformity of the plume (x2, x4), whereas others are explained by instrumental flaws (x1). A better characterization of the plume would strengthen the interpretation.

Regarding the particle sizing, a FMAG was used to produce mono-dispersed droplets (in the range of 15 to 72  $\mu$ m in diameter). For the size calibration experiments, droplet diameters of 20.13, 39.35 and 57.55  $\mu$ m were chosen. Would it be possible to produce size calibration experiments with smaller sizes (e.g. by using a Vibrating Orifice Aerosol Generator (VOAG); or PMMA/PSL spheres with atomizer or fluidized bed)? Previous studies have shown that sizing uncertainties are largest for the smallest particles, so I would recommend performing sizing experiments for smaller sizes and if possible include additional instruments for better validation. Furthermore, cloud droplets are generally smaller than 20  $\mu$ m. For example, the cloud droplet size observed in Sect. 5 all lie below the smallest calibration size applied in this study (20.13  $\mu$ m).

**Specific comments**

1. Page 1 line 19: You mention that researchers have to tackle with intrinsic difficulties

when using in situ and remote sensing observations. You could consider giving some examples describing the main challenges of in situ and remote sensing observations.

2. Page 2 line 44: Holographic systems have also been applied on balloon-borne platforms; e.g. HoloBalloon Ramelli et al. 2020, https://amt.copernicus.org/articles/13/925/2020/

3. Page 2 line 48: Here you compare the sampling volume and frame rate of holographic and shadowgraph instruments. I would recommend to compare the sample volume rate as this is more meaningful (e.g. 15 cm3\*5 fps = 75 cm3 s-1; 0.04 cm3\*400 fps = 16 cm3 s-1).

4. Page 2 line 50: You write that the shadowgraph technique did not gain common use in cloud measurements. This is misleading, as CPI are frequently used on aircrafts. Here are some examples:

Lawson et al. 2001: https://agupubs.onlinelibrary.wiley.com/doi/pdf/10.1029/2000JD900789

Lawson et al. 2010 : https://agupubs.onlinelibrary.wiley.com/doi/pdf/10.1029/2009JD013017

Stith et al. 2002: https://journals.ametsoc.org/jamc/article/41/2/97/16088

Woods et al. 2018: https://agupubs.onlinelibrary.wiley.com/doi/pdf/10.1029/2017JD028068

5. Page 7 line 146: In Fig. 1 the flow was vertically aligned. Did you use a different flow direction for the experiments? Please specify.

6. Page 9 Fig. 4: On page 7 line 142 you state that the poly-dispersed water droplets in the stream were in the size range 2-20  $\mu$ m. Is there a way to verify this? Why do you observe a lot of droplets larger than 20  $\mu$ m in Fig. 4?

7. Page 14 line 280: I only see the gradual decrease from the center to the sides for x1 in the horizontal direction. It seems like you have more particles on the left side compared to the right side (maximum is not at 0  $\mu$ m). Is this relating to the non-uniformity of the plume or to instrumental flaws?

8. Page 14 line 284: You speculate that the decrease in concentration in vertical direction for x1 is due to instrumental reasons (e.g. non-uniform illumination). Can you verify/quantify that based on the images? The shadow images for x1 in Fig. 12 don't show indications for non-uniform illumination. Please comment on that.

9. Page 14 line 292: You explain that the axial dependence is more difficult to evaluate. Fig. 8.c shows a sharp decrease of the concentration with increasing z-distance, which you attribute to the miscounting of the smaller droplets. Would it be possible to perform a similar experiment as in Sect.4 where you generate a mono-dispersed particle distribution? Or are the concentrations produced by the FMAG to small?

10. Page 16 line 335: For the size calibration experiments, you produced droplet diameters of 20.13, 39.35 and 57.55  $\mu$ m. Can you explain how these diameters were selected?

11. Page 18 line 353: You state that the left-side skewness of the tails in Fig. 11 implies partial evaporation. I only see a left-skewed tail for x2 and x1 for the diameters 39.25 and 57.55  $\mu$ m, but not for x4 and the smaller size (20.13  $\mu$ m). I would expect that the evaporation effect is largest for smaller particles, but this does not seem to be the case. How can you explain this pattern?

12. Page 21 Fig. 14: Why do you have multiple times the same symbol in a, b and c? Are these different experiments?

13. Page 22 Fig. 15: Were other cloud probes deployed at the measurement site? On page 20 line 381 you say "After laboratory tests, the shadowgraph VisiSize D30 has been used for the first time to measure droplets in atmospheric clouds [...] to compare it with other probes already in service in cloud physics studies [...]". If other cloud probes were deployed at the same time, I would add the size distribution of additional probes in Fig. 15 for validation of the VisiSize D30 shadowgraph system. Alternatively, I would recommend performing a comparison campaign with other cloud probes in the future.

**Technical comments**

1. Page 5 line 93: I would suggest to write "[. . . ] a method compensating for the effects  $[.\,.\,.]$  "

2. Page 6 Fig. 2: Consider choosing different colors for Th and Tp for better distinction.

- 3. Page 6 line 120: 'z' should be written in italic
- 4. Page 8 line 165: 'z' should be written in italic
- 5. Page 8 line 167: 'z' should be written in italic
- 6. Page 15 line 308: capitalize 'D' in "2-dimensional (z, D) maps"

7. Page 19 Fig. 12: Consider adding different lines for single droplet/double collision/triple collision similar as in Figure 13 or at least of the FMAG diameter.

---

## Author Comment (AC1) · 16 Dec 2020

**Authors' Response to the Anonymous Referee #1**

Jakub L. Nowak, Moein Mohammadi, Szymon P. Malinowski

We are grateful to the Referee #1 for the insightful comments and suggestions on our manuscript. We respond to them in detail below. The original review is given in black, our anwers in blue. The responses mention also specific corrections which were applied to the manuscript.

**General comments**

It is challenging to characterize the sample volume of the shadowgraph system (Sect. 3) without accurate information regarding the properties of the plume (e.g. uniformity, size distribution of poly-dispersed droplets). Uncertainties in the calibration method can have large uncertainties on the droplet number concentration and size distribution. The assumptions and speculations in Sect. 3 are all justified, but some uncertainties remain. For example, the different lens systems show different dependencies of the number concentration on the horizontal, vertical and axial position (Fig. 8), some of which are attributed to the non-uniformity of the plume (x2, x4), whereas others are explained by instrumental flaws (x1). A better characterization of the plume would strengthen the interpretation.

We agree with the referee that the better characterization of the plume would strengthen the interpretation, i.e. allow to distinguish the effect of its non-uniformity from instrumental flaws. However, we were not able to do this with the available instrumentation. Therefore, we point out in the text that some particular conclusions cannot be delivered with high confidence based on our experiment.

The most important correction we developed involves limitation of the depth of field  $z_{95}$  in Eq. (5). It was first derived analytically and the experiment proved to confirm its validity. It did not serve to derive the calibration, i.e. none of the coefficients used in data processing was fitted to the collected data. The calibration constants  $a_1, \ldots, a_6$  were prior provided by the manufacturer and resulted from the calibration performed with the Patterson globe targets (Kashdan et al., 2003).

Concerning dependency of the number concentration on the position illustrated in Fig. 8, we do not claim that in case of lens setting x2 and x4 the effect is entirely due to the non-uniformity of the plume. What we mean is that such dependency might have been caused by the plume properties, hence we are not entitled to blame instrumental issues on the ground of our results. Lens setting x1 features such a large difference that, in our opinion, it is hardly possible to explain it with the properties of the plume only.

Regarding the particle sizing, a FMAG was used to produce mono-dispersed droplets (in the range of 15 to 72  $\mu$ m in diameter). For the size calibration experiments, droplet diameters of 20.13, 39.35 and 57.55  $\mu$ m were chosen. Would it be possible

to produce size calibration experiments with smaller sizes (e.g. by using a Vibrating Orifice AerosolGenerator (VOAG); or PMMA/PSL spheres with atomizer or fluidized bed)? Previous studies have shown that sizing uncertainties are largest for the smallest particles, so I would recommend performing sizing experiments for smaller sizes and if possible include additional instruments for better validation. Furthermore, cloud droplets are generally smaller than 20 µm. For example, the cloud droplet size observed in Sect. 5 all lie below the smallest calibration size applied in this study (20.13 µm).

We are aware of this limitation. Naturally, it would be of advantage to verify the sizing of smaller droplets as we also expect largest uncertainties for the smallest particles. However, the instrument which was at our disposal (FMAG) can reach down to 17  $\mu$ m in diameter using ultra purified water (UPW) as the fluid (Duan et al., 2016). In the course of the experiment, we tried to generate droplets smaller than 20  $\mu$ m but this resulted in quite a wide spread of the diameters. In FMAG, the size is controlled by the liquid flow rate and the vibration frequency. To obtain small sizes, high vibration frequency is necessary (e.g. 200 Hz for the diameter of 17  $\mu$ m, see Table 2 in Duan et al. (2016), which probably results in less-accurate breakage of the fluid stream into droplets and more likely collisions. The smallest diameter we used in this study (20.13  $\mu$ m) was chosen as a compromise to avoid these effects.

We plan to develop a special calibration target composed of dots of known diameters, similar to that applied by Kashdan et al. (2003) but in the size range below  $20 \,\mu\text{m}$ . Unlike the droplets, the position of the target in the sampling volume would be under control.

**Specific comments**

1. Page 1 line 19: You mention that researchers have to tackle intrinsic difficulties when using in situ and remote sensing observations. You could consider giving some examples describing the main challenges of in situ and remote sensing observations.

We listed a few examples for each of the two approaches:

Researchers employing both strategies tackle with intrinsic difficulties. For instance, in-situ methods often face the dependence of the sample volume on particle size or air flow velocity, nonlinearity of Mie scattering intensity with respect to droplet size, aerodynamic effects related to the flow around or inside the instrument or aircraft, harsh conditions (incl. icing, wetting, temperature changes), necessity for handling large datasets or instantaneous data processing. Remote sensing provides the information with limited spatial resolution, hence microphysical properties represents only the average or integral over relatively large volumes which might be too simplistic to characterize inhomogeneous or multilayered cloud fields. On top of that, the retrievals are often dependent on the assumptions of specific size distribution or specific vertical structure of the atmosphere.

 Page 2 line 44: Holographic systems have also been applied on balloon-borne platforms; e.g. HoloBalloon Ramelli et al. 2020, https://amt.copernicus.org/articles/13/925/2020/

We included this study in the short review of cloud droplet sizing instruments given in the introduction. We also updated the citation concerning the HALOHolo instrument. Together with Schlenczek et al. (2017) who actually performed ground-based measurements, we cited Schlenczek (2017) (PhD thesis) and Lloyd et al. (2020).

3. Page 2 line 48: Here you compare the sampling volume and frame rate of holographic and shadowgraph instruments. I would recommend to compare the sample volume rate as this is more meaningful (e.g. 15 cm3\*5 fps = 75 cm3 s-1; 0.04 cm3\*400 fps = 16 cm3 s-1).

We agree, those values were mentioned in the revised text. In case of holography we used the value of 6 fps in this estimation as this is the true frame rate of e.g. HALOHolo and HoloGondel. Additionally, we cited the recent advancements with regard to the sample volume rate achieved in HoloBallon by Ramelli et al. (2020).

4. Page 2 line 50: You write that the shadowgraph technique did not gain common use in cloud measurements. This is misleading, as CPI is frequently used on aircrafts. Here are some examples: Lawson et al. 2001: https://agupubs.onlinelibrary.wiley.com/doi/pdf/10.1029/2000JD900789 Lawson et al. 2010: https://agupubs.onlinelibrary.wiley.com/doi/pdf/10.1029/2009JD013017 Stith et al. 2002: https://journals.ametsoc.org/jamc/article/41/2/97/16088 Woods et al. 2018: https://agupubs.onlinelibrary.wiley.com/doi/pdf/10.1029/2017JD028068

We did not formulate the point clearly. What we actually meant is that shadowgraphy is not common in cloud droplet measurements in comparison to other methods. Additionally, we cited the first from the papers listed because it describes the capabilities of the CPI, and the last one to give the most recent specification on the size range measured by this instrument.

The corrected sentence reads:

Nevertheless, despite both its simplicity and many insightful laboratory experiments, e.g. concerning droplet collisions (Bordás et al., 2013; Bewley et al., 2013), shadowgraphy is not the first choice method in cloud droplet measurements.

5. Page 7 line 146: In Fig. 1 the flow was vertically aligned. Did you use a different flow direction for the experiments? Please specify.

Fig. 1. illustrates the components of the setup. The photograph was taken during another short test performed with the same instruments but different mounting of the pipe. It is neither analyzed nor discussed in the paper. During the measurements described in the paper, the outflow from the pipe was horizontally aligned. Unfortunately, no picture of this particular configuration is available.

Appropriate explanation was added to the text in sec. 3.1:

The flow velocity was estimated to be of the order of  $10 \text{ cm s}^{-1}$  and the direction of the flow was aligned horizontally from left to right (i.e. the direction of the pipe exit was perpendicular to what is shown in Fig. 1).

6. Page 9 Fig. 4: On page 7 line 142 you state that the poly-dispersed water droplets in the stream were in the size range 2-20 μm. Is there a way to verify this? Why do you observe a lot of droplets larger than 20 μm in Fig. 4?

Unfortunately, we do not have another droplet sizing instrument. The given approximate range comes from the works of P. Korczyk and others (Korczyk, 2008; Korczyk et al., 2012) who used the same source of droplets, i.e. an ultrasonic humidifier. They measured the size spectrum by collecting sedimenting droplets on a glass plate covered with an oil film to prevent evaporation and analyzing the microscopic photographs. Despite the same source, the delivery system and ambient thermodynamic conditions are somewhat different in our case, which might result in small changes in the actual droplet size distributions due to possible collisions and evaporation. Note also that we collected much larger statistics (i.e. total number of counts) than was feasible with their method, hence very infrequent large droplets are better represented in our dataset.

We added the explanation in the text:

A dense stream of poly-disperse water droplets was generated with the use of an ultrasonic humidifier, the same as in the study of Korczyk et al. (2012) who measured the droplet size to be mostly in the range of  $2-20 \,\mu\text{m}$  in diameter (Fig. 1 there). Differences in delivery method and in ambient conditions could result in a little different spectrum.

7. Page 14 line 280: I only see the gradual decrease from the center to the sides for x1 in the horizontal direction. It seems like you have more particles on the left side compared to the right side (maximum is not at 0 µm). Is this relating to the non-uniformity of the plume or to instrumental flaws?

Our interpretation of the logarithmic plot given in the text was too simplistic. We changed the quantity presented in panels (a), (b) of Fig. 8 by normalizing  $N_V(x)$  or  $N_V(y)$  by the mean concentration to show the relative changes. After this correction it is clear the maximum is located left from the center. We suppose this might be related to the non-uniformity of the plume as well as the instrumental flaws. Therefore, we cannot claim the specific instrumental issues based on the results.

**The corrected description reads:**

Figure 8 in panels (a) and (b) shows normalized droplet concentration  $N_V(x)/N_V$  and  $N_V(y)/N_V$  suitably integrated over size and over other dimensions, divided by the total concentration in order to highlight the relative dependence on position inside the sample volume. The values mostly decrease gradually from the maximum (located left from the center) to the sides in case of horizontal direction and from the bottom to the top in case of vertical. The relative differences, except from very close to the edges, are of the order of 10 %. They are small enough to be possibly caused by the non-uniformities of the plume.

Figure 8 corrected. Dependence of droplet concentration on (a) horizontal, (b) vertical and (c) axial position.

8. Page 14 line 284: You speculate that the decrease in concentration in vertical direction for x1 is due to instrumental reasons (e.g. non-uniform illumination). Can you verify/quantify that based on the images? The shadow images for x1 in Fig. 12 don't show indications for non-uniform illumination. Please comment on that.

The VisiSize software has an option of background normalization. The user can select one image which serves as a background and all the further frames collected in the measurement are divided by it. Such an option was used in the detection experiment described in sec. 3 (Fig. 3). We speculate the background normalization ensures proper performance of the thresholds ( $T_h$  and  $T_p$ ) but does not substantially improve the signal-to-noise ratio in originally darker (weakly illuminated) regions. There, only a small part of the grayscale levels (256) is used.

We cannot verify/quantify this issue directly for our detection experiment because in the course of long runs, the images are not saved but instantly processed to derive particle statistics (see the list of them in Table 2). If one prefers to keep the captured images, the duration of an uninterrupted measurement is limited by the available computer RAM which in our particular configuration meant about 2000 frames (corresponding to about 66 seconds).

In the series shown in Fig. 12 some gradient of brightness is visible in case of lens setting x1, for instance the right edge seems to be slightly darker than the center and the left. The orientation of the gradient is then different than in the detection experiment (sec. 3) which might result from manual position adjustment. We recall our observation that the satisfactory adjustment for this lens setting is rather difficult to achieve. Importantly, most of the droplets measured during the sizing experiment (sec. 4) were close to the central part of the SV. Therefore, our findings are not affected by the illumination non-uniformity.

9. Page 14 line 292: You explain that the axial dependence is more difficult to evaluate. Fig. 8.c shows a sharp decrease of the concentration with increasing z-distance, which you attribute to the miscounting of the smaller droplets. Would it be possible to perform a similar experiment as in Sect. 4 where you generate a mono-dispersed particle distribution? Or are the concentrations produced by the FMAG too small?

The concentrations produced by the FMAG are indeed much smaller  $(0.3-3 \text{ mm}^{-3})$  in comparison to the dense plume in the first experiment. We speculate the exact value depends non-trivially on the settings of the FMAG and the conditions on the way from the nozzle exit to the sample volume of the shadowgraph (flow properties, evaporation, collisions). Moreover, the generated stream of droplets is quite localized which implies non-random probing of the sample volume, including the axial direction. The essential component of the detection experiment with the dense plume (sec. 3) was the close-to-uniform filling of the SV. In our opinion, appropriate study of detection properties involving FMAG-generated droplets would require precise control of the position and orientation of the droplet stream with respect to the focal plane of the shadowgraph which seems to be challenging.

 Page 16 line 335: For the size calibration experiments, you produced droplet diameters of 20.13, 39.35 and 57.55 μm. Can you explain how these diameters were selected?

We intended to select 3 different sizes covering the range which is available with the particular FMAG generator at our disposal using UPW (17-69  $\mu$ m, Duan et al. (2016)). As explained in our reply to the general comment above, the smallest diameter (20.13  $\mu$ m) was chosen as a compromise in order to avoid spectrum broadening. Such effect is most probably related to the high vibration frequency which together with the liquid flow rate controls the output size.

We added a short explanation of the choice of the smallest size in the text:

The smallest diameter was chosen to ensure a relatively narrow spectrum as we observed significant broadening for high vibration frequency which is necessary to generate yet smaller sizes. Presumably, this results in less accurate breakage of the fluid stream or more frequent collisions.

11. Page 18 line 353: You state that the left-side skewness of the tails in Fig. 11 implies partial evaporation. I only see a left-skewed tail for x2 and x1 for the diameters 39.25 and 57.55 μm, but not for x4 and the smaller size (20.13 μm). I would expect that the evaporation effect is largest for smaller particles, but this does not seem to be the case. How can you explain this pattern?

In our opinion, the observation made by the reviewer might be related to the size of the sample volume which increases with effective pixel size (decreases with magnification) as well as with the particle size (i.e. from the top-left to the bottom-right panel in Fig. 12). Other processes which were not controlled in our experiment might have also contributed to the observed results, e.g. ambient air properties, velocity of the droplets or some interactions among them.

Our speculation was explained in the text:

Left-side skewness of the tails suggests partial evaporation. Although in general the effect of evaporation is expected to be more significant for small droplets, the skewness is evident for 39.25 and 57.55  $\mu$ m measured with the lens settings x1 and x2. We speculate it might be related to the size of the sample volume which increases with the effective pixel size (decreases with magnification) as well as with the particle size (i.e. from the top-left to the bottom-right panel in Fig. 12). The position of the nozzle exit was adjusted so that the

center of the FMAG-generated droplet stream is as close as possible to the focal plane of the shadowgraph. We expect the droplets more distant from the central axis of the stream to be more likely partially evaporated because of the longer travel and exposure to the dry air blown from the area around the nozzle. Those can be detected in case of considerable sample volume but not in case of smaller SV. Importantly, this is only one of the effects which could have contributed to the observed result together with the ambient air properties, velocity of the droplets or some interactions among them.

- 12. Page 21 Fig. 14: Why do you have multiple times the same symbol in a, b and c? Are these different experiments? Yes, there were multiple measurement rounds performed with the same settings. Each symbol represents the mean diameter or the 1st-peak diameter obtained in a single measurement. The caption of the figure was updated accordingly.
- 13. Page 22 Fig. 15: Were other cloud probes deployed at the measurement site? On page 20 line 381 you say "After laboratory tests, the shadowgraph VisiSize D30 has been used for the first time to measure droplets in atmospheric clouds [...] to compare it with other probes already in service in cloud physics studies [...]". If other cloud probes were deployed at the same time, I would add the size distribution of additional probes in Fig. 15 for validation of the VisiSize D30 shadowgraph system. Alternatively, I would recommend performing a comparison campaign with other cloud probes in the future.

During two observational periods in July and August 2019, the size distribution was measured also by the Phase Doppler Interferometer (PDI). The two instruments, intentionally located side-by-side were not simultaneously operational in all the runs but we have data from overlapping periods. This data is still subject to processing and detailed analysis. In case of the PDI, the sample volume depends non-trivially not only on droplet size but also on the wind velocity, highly variable in the turbulent experimental conditions. Therefore, we are working on proper systematic comparison which considers those issues and shows some limitations of ground-based measurements with the instruments developed for high true air speeds . We intend to describe the details and submit it to the AMT as soon as the analysis and interpretation is finished.

**Technical comments**

- 1. Page 5 line 93: I would suggest to write "[...] a method compensating for the effects[...]"
- 2. Page 6 Fig. 2: Consider choosing different colors for Th and Tp for better distinction.
- 3. Page 6 line 120: 'z' should be written in italic.
- 4. Page 8 line 165: 'z' should be written in italic.
- 5. Page 8 line 167: 'z' should be written in italic.

- 6. Page 15 line 308: capitalize 'D' in "2-dimensional (z, D) maps"
- 7. Page 19 Fig. 12: Consider adding different lines for single droplet/double collision/triple collision similar as in Figure 13 or at least of the FMAG diameter.

We agree with the technical comments and applied the specific corrections according to the reviewer's suggestions.

**References**

- Bewley, G. P., Saw, E. W., and Bodenschatz, E.: Observation of the sling effect, New Journal of Physics, 15, https://doi.org/10.1088/1367-2630/15/8/083051, 2013.
- Bordás, R., Roloff, C., Thévenin, D., and Shaw, R. A.: Experimental determination of droplet collision rates in turbulence, New Journal of Physics, 15, 045 010, https://doi.org/10.1088/1367-2630/15/4/045010, 2013.
- Duan, H., Romay, F. J., Li, C., Naqwi, A., Deng, W., and Liu, B. Y. H.: Generation of monodisperse aerosols by combining aerodynamic flowfocusing and mechanical perturbation, Aerosol Science and Technology, 50, 17–25, https://doi.org/10.1080/02786826.2015.1123213, 2016.
- Kashdan, J. T., Shrimpton, J. S., and Whybrew, A.: Two-Phase Flow Characterization by Automated Digital Image Analysis. Part 1: Fundamental Principles and Calibration of the Technique, Particle & Particle Systems Characterization, 20, 387–397, https://doi.org/10.1002/ppsc.200300897, 2003.
- Korczyk, P. M.: Drobnoskalowa turbulencja w procesie mieszania chmury z otoczeniem model laboratoryjny, Ph.D. thesis, Polish Academy of Sciences, 2008.
- Korczyk, P. M., Kowalewski, T. A., and Malinowski, S. P.: Turbulent mixing of clouds with the environment: Small scale two phase evaporating flow investigated in a laboratory by particle image velocimetry, Physica D: Nonlinear Phenomena, 241, 288–296, https://doi.org/10.1016/j.physd.2011.11.003, 2012.
- Lloyd, G., Choularton, T., Bower, K., Crosier, J., Gallagher, M., Flynn, M., Dorsey, J., Liu, D., Taylor, J. W., Schlenczek, O., Fugal, J., Borrmann, S., Cotton, R., Field, P., and Blyth, A.: Small ice particles at slightly supercooled temperatures in tropical maritime convection, Atmospheric Chemistry and Physics, 20, 3895–3904, https://doi.org/10.5194/acp-20-3895-2020, 2020.
- Ramelli, F., Beck, A., Henneberger, J., and Lohmann, U.: Using a holographic imager on a tethered balloon system for microphysical observations of boundary layer clouds, Atmospheric Measurement Techniques, 13, 925–939, https://doi.org/10.5194/amt-13-925-2020, 2020.
- Schlenczek, O.: Airborne and ground-based holographic measurement of hydrometeors in liquid-phase, mixed-phase and ice clouds, Phd thesis, Johannes Gutenberg University in Mainz, https://doi.org/10.25358/openscience-4124, 2017.
- Schlenczek, O., Fugal, J. P., Lloyd, G., Bower, K. N., Choularton, T. W., Flynn, M., Crosier, J., and Borrmann, S.: Microphysical Properties of Ice Crystal Precipitation and Surface-Generated Ice Crystals in a High Alpine Environment in Switzerland, Journal of Applied Meteorology and Climatology, 56, 433–453, https://doi.org/10.1175/JAMC-D-16-0060.1, 2017.

---

## Author Comment (AC2) · 16 Dec 2020

**Authors' Response to the Anonymous Referee #2**

Jakub L. Nowak, Moein Mohammadi, Szymon P. Malinowski

We are grateful to the Referee #2 for the insightful comments and suggestions on our manuscript. We respond to them in detail below. The original review is given in black, our anwers in blue. The responses mention also specific corrections which were applied to the manuscript.

**General comments**

Would it be possible to use the shadowgraph system to obtain some 3D information about the position of the cloud droplets?

Unfortunately, the diameter-dependent sample volume is generally too small to obtain the information about the 3D arrangement of the droplets in a typical atmospheric cloud. For instance, in Fig. 5. one can see the DOF for the droplet of a diameter 20  $\mu$ m is about 200  $\mu$ m (twice the  $z_{max}$ ). Multiplied by the FOV (see Table 1) it results in the SV of about 5.90, 1.50 and 0.35 mm3, for magnifications x1, x2 and x4, respectively. Even if the concentration was 1000 cm-3 and all the droplets had 20  $\mu$ m, we would expect on average around 5.9, 1.5 or 0.35 counts per frame, respectively. For smaller diameters, the DOF and hence the SV as well as the expected counts per frame are accordingly much smaller. For the field measurement described in sec. 5, the concentrations were estimated to about 928 cm-3 and 798 cm-3. During the first round with magnification x2, there were on average only 0.7 counts per frame. During the second round with magnification x4, there were on average only 0.08 counts per frame.

Another issue might be the shape of the SV resembling more a slice than a cube as the z-dimension is much smaller than x and y. On top of that, only the absolute value of z coordinate of a droplet is estimated. It is unknown, on which side of the focal plane the object is.

Could you give any estimate for the largest detectable number concentration? In case of large number concentrations, for example, it might be possible that droplets "hide" one after the other.

Due to the small slice-shaped SV, the problem of the largest detectable concentration is not relevant for typical atmospheric clouds. Similarly to the previous question, we may consider mono-dispersed droplets of a diameter 20  $\mu$ m and concentration 1000 cm-3. Then, the expected number in the single cylinder of a diameter 40  $\mu$ m and the height equal to the DOF (around 200  $\mu$ m) is only 2.5  $\cdot 10^{-4}$ . For smaller droplets, it would be accordingly smaller. Therefore, the coincidence of two droplets at similar (*x*, *y*) position seems highly improbable.

In the field measurements, there was on average even below 1 count per frame. The concentration in the plume used during the laboratory experiment was about two orders of magnitude larger which gave 18-54 counts per frame (depending on magnification). In the limited number of the inspected test images we did not spot coincidences. However, in the course of the longer measurement the images are not stored but processed in real time (see sec. 2.1) which makes the problem of coincidences hardly possible to verify afterwards.

On the other hand, what we observed during the laboratory experiment is that the dense plume filling the considerable space between the camera and the laser but outside the SV can influence the average brightness of the image. The thresholds implemented in the software adapt to the average brightness but still such shadow diminishes signal-to-noise ratio of individual particles as discussed for the magnification x1 in sec. 3.5. In our opinion, the effect is not significant in atmospheric clouds due to smaller concentration, hence limited scattering.

Concerning the application of the shadowgraph system in the atmospheric cloud (Section 5): Has there been any other cloud probe which measured the size distribution at the same time as the shadowgraph system? If so, could you present a comparison of the obtained droplet size distributions and its moments, respectively? If not, I would suggest, for a future study, to have a comparison to another cloud probe.

During two observational periods in July and August 2019, the size distribution was measured also by the Phase Doppler Interferometer (PDI). The two instruments, intentionally located side-by-side were not simultaneously operational in all the runs but we have data from overlapping periods. This data is still subject to processing and detailed analysis. In case of the PDI, the sample volume depends non-trivially not only on droplet size but also on the wind velocity, highly variable in the turbulent experimental conditions. Therefore, we are working on proper systematic comparison which considers those issues and shows some limitations of ground-base measurements with the instruments developed for high true air speeds. We intend to describe the details and submit it to the AMT as soon as the analysis and interpretation is finished.

**Specific comments**

1. Page 7, line 145-146: Looking on Fig. 1, I would think that the direction of the droplet flow was vertically aligned.

Fig. 1. illustrates the components of the setup. The photograph was taken during another short test performed with the same instruments but different mounting of the pipe. It is neither analyzed nor discussed in the paper. During the measurements described in the paper, the outflow from the pipe was horizontally aligned. Unfortunately, no picture of this particular configuration is available.

Appropriate explanation was added to the text in sec. 3.1:

The flow velocity was estimated to be of the order of  $10 \text{ cm s}^{-1}$  and the direction of the flow was aligned horizontally from left to right (i.e. the direction of the pipe exit was perpendicular to what is shown in Fig. 1).

2. Fig. 5c: For diameters larger than 30  $\mu$ m,  $z_{max}$  deviates significantly from the analytical approximation  $z_{95}$ . Do you have any explanation for that?

The discrepancy relates to the poor statistics of large droplets. Each point in the plot represents one diameter bin and shows the distance from the focal plane corresponding to the furthest detected droplet from that bin. Diameters larger than 30  $\mu$ m are very scarce in the plume (see the size distribution in Fig. 7), so the number of detections is also small and they do not fill equally the entire sample volume. Therefore, it can happen that all of them randomly appear at the positions closer to the focal plane than the maximum possible distance  $z_{95}$ . For instance, in the case of the measurement with the magnification x4 presented in Fig. 5c, there were only 3 counts in the last bin and only 15 in the one before last. The sample volume decreases strongly with magnification implying the limited chance of detecting infrequent large droplets which explains why the effect is most pronounced for x4.

We added a short comment to the text:

The discrepancy observed for largest sizes is related to the poor statistics, i.e. small number of counts of large droplets which are infrequent in the measured plume.

3. Fig. 7 and Fig. 15 and the corresponding text: In Fig. 7 only DSDs applying methods"def" and "ind" are shown, for given reasons. However, later in Fig. 15, DSDs for "def", "cor" and "ind" are shown. Wouldn't it make more sense, to show DSDs for methods"def", "cor" and "ind" in Fig. 7 and explain that there are only small differences between methods "cor" and "ind" and then only show DSDs for "def" and "ind" in Fig. 15?

We added the missing method to Fig. 7 and decided to keep all of them in Fig. 15 for consistency. The caption of Fig. 7 was modified accordingly.

4. Page 14, line 280: This gradual decrease from the center to the sides is obvious for x1, but not really for x2 and x4. Could you please comment on that?

Our interpretation of the logarithmic plot given in the text was too simplistic. We changed the quantity presented in panels (a), (b) of Fig. 8 by normalizing  $N_V(x)$  or  $N_V(y)$  by the mean concentration to show the relative changes. After this correction it is clear the maximum is located left from the center.

The corrected description reads:

Figure 8 in panels (a) and (b) shows normalized droplet concentration  $N_V(x)/N_V$  and  $N_V(y)/N_V$  suitably integrated over size and over other dimensions, divided by the total concentration in order to highlight the relative dependence on position inside the sample volume. The values mostly decrease gradually from the maximum (located left from the center) to the sides in case of horizontal direction and from the bottom to the top in case of vertical. The relative differences, except from very close to the edges, are of the order of 10 %. They are small enough to be possibly caused by the non-uniformities of the plume.

Figure 8 corrected. Dependence of droplet concentration on (a) horizontal, (b) vertical and (c) axial position.

5. Page 14, line 284-291: I think the speculation given here is reasonable. However, could this also be true for x2 and x4? But here more (smaller) droplets are detected, both in a smaller FOV compared to x1 which compensates this feature?

As far as we understand, the reviewer suggests the suspected y-dependent concentration concerns large droplets regardless of the magnification setting used. This effect is then supposedly not clearly visible for x2 and x4 because of the dominant number of small droplets detected in a more uniform manner.

We verified this claim to be true. The concentration  $N_V(y)$  of the droplets larger than 12 µm only is indeed more ydependent than the same quantity for the droplets smaller than 12 µm, in case of both magnifications x2 and x4. We cannot exclude the contribution of gravity sorting to that effect. However, the relative decrease of the  $N_V(y)$  of large droplets from the bottom to the top is significantly smaller than over the same distance in case of the magnification x1.

The exact influence of the non-uniform illumination on the results of DSD and DNC is hardly possible to be estimated with the available data. We emphasize the importance of precise alignment of the laser and camera lens positions. This is done manually, so the positioning can differ slightly between the experiments. As stated in the paper, achieving close to uniform illumination is much easier for magnifications x2 and x4 than x1 because of the smaller FOV to be fitted into the same laser beam.

We modified the text:

The y-dependence of the concentration is more pronounced for large droplets above 12  $\mu$ m (not shown) which might be the influence of gravity sorting. However, in case of the series recorded with lens setting x1, the concentration falls with height y by a factor of more than 10 from the bottom to the top of the FOV. We speculate this effect is of instrumental origin, as the difference is rather too large (almost exponential) and the timescale too short to allow for the explanation only by gravity sorting of the droplets in the plume.

 Page 15, Line 318: Could you please provide a size range where you would not use x1. As later said, x1 makes sense for larger droplets in the drizzle and rain size range. Please make clearer under which circumstances you would avoid using x1, and vice versa.

We estimated the minimum diameter for uniform detection to 12  $\mu$ m so the magnification x1 is not recommended to measure any droplets smaller than that. Another limit, related to the issue of illumination, is not so precise because the image quality depends to some extent on the accuracy of manual alignment. In order to avoid poor signal-to-noise ratio we would suggest, as a rule of thumb, a diameter an order of magnitude larger than the effective pixel size which results in around 40  $\mu$ m. On the other hand, the objects need to fit into the FOV, so the largest measured size should be a few times smaller than the shorter dimension (4.1 mm). Eventually, we expect the setting x1 to perform well in the range of roughly 40-400  $\mu$ m. So far, we have not collected much data on drizzle to confirm this expectation experimentally. Nevertheless, in the sizing experiment (sec. 4) we had indeed no issues for 39.35  $\mu$ m droplets but still some inaccuracies for 20.13  $\mu$ m.

We included this information in the text:

As a consequence, the authors discourage using lens setting x1 in further studies of cloud droplets. The large minimum particle size for uniform detection reported earlier ( $\sim$ 12 µm) makes this option of limited utility anyway. Instead, x1 can possibly serve well for the measurements of drizzle drops. In order to avoid the illumination and signal-to-noise issue described above, we would suggest, as a rule of thumb, the lower limit an order of magnitude larger than the effective pixel size (3.69 µm). On the other hand, the objects need to fit into the FOV, so the largest measured size should be a few times smaller than the shorter dimension (4.1 mm) which eventually results in the conservative range of roughly 40-400 µm. Notwithstanding, so far insufficient data on drizzle has been collected to confirm this expectation experimentally.

7. Fig. 9 and respective text: Does the discrepancy in the first and second z-bin has any consequences on the calculation of the DSD and DNC?

We consider the question relevant however cannot estimate the effect quantitatively based on the available data. In order to investigate this issue we increased the number of z-bins from 20 to 80 which implied the changes in Fig. 8c and Fig. 9. This operation allowed to identify a lower limit on the distance z and the probable reason for the discrepancy observed earlier - diffraction around the objects close to the focal plane. In the paragraphs added to the text we discuss the mechanism and speculate the influence on the DSD and the DNC within the valid size range is rather minor. In our opinion, one solution to systematically eliminate the problem would be to add the term representing diffraction to Eq. (1). This requires repeating the calibration procedure to obtain the correct new values of  $a_1$  and  $a_2$ . At best, also implementing the corrected equation in the software processing routine.

We corrected the interpretation of Fig. 8c:

Figure 9 corrected. Variability of droplet detection properties in size-space domain for lens setting (a) x1, (b) x2, (c) x4.

Interestingly, the first z ranges contain a much smaller number of droplets than the maximum located further from the focal plane, regardless of the lens setting.

We added the paragraphs discussing the issue:

On the other side, the minimum distance  $z_{min}$  for a given diameter is well approximated by assuming the halo area in the focal plane equal to the diffraction term ( $A_h = \pi D a_5$ ), analogously to Fig. 4, and solving Eq. (1) for z (blue line).

(...)

In experimental runs with higher magnifications, i.e. lens setting x2 and x4, the (z, D) map features the decrease of concentration with z from the maximum located a bit above  $z_{min}$ . The same effect was noticed in Fig. 8 panel (c). Most probably, it relates to the diffraction which is not included in the modeled dependence of the halo area  $A_h$  on the diameter D and distance z in Eq. (1). Namely, the equation is not correct in the limit of small z because it implies the object in the focal plane (z = 0) should be ideally sharp  $(A_h = 0)$ . Consequently, in this limit the calculated z position is overestimated with respect to the true one. The counts representing droplets standing very close to the focal plane are shifted to the further z-bins in Fig. 9. The extent of this shift is probably not constant but decreases with the true z. Hence, the counts cumulate at some point above  $z_{min}$  creating a maximum in  $N_V(z)$ . We expect the shift to decrease because the calibration constants  $a_1$  and  $a_2$  were fitted by the manufacturer in the procedure resembling Kashdan et al. (2003) so that the Eq. (1) performs satisfactorily in the range of defocus distances and particle diameters typical for industrious applications, i.e. a bit larger than analysed here. Therefore, the estimated z should approach the true one with increasing defocus and droplet diameter.

The shift of the estimated z positions with respect to the true ones is most pronounced in case of the small distances from the focal plane. Importantly, it should have no effect on the accuracy of the sample volume calculation, hence the DNC and the DSD, as long as the  $z_{95}$  is not overestimated but represents the

true distance at which the droplets are no longer counted. We expect this condition to be met if the largest possible defocus  $z_{95}$  is significantly higher than the smallest  $z_{min}$ . For instance, they differ by a factor of two for diameters larger than 8.0, 6.3, 4.8 µm in case of lens settings x1, x2 and x4, respectively. Those values are close to the minimum diameter for uniform detection estimated in sec. 3.4. It implies the influence on the DSD and the DNC within the valid size range is probably minor. However, we cannot quantify it with high confidence based on the available data.

**Minor comments**

- 1. Page 2, Line 50: I would suggest to write either "shadowgraphy" or "the shadowgraph technique".
- 2. Page 3, line 70: Please add "the" at the beginning of the sentence: "The two main parts[...]".
- 3. Page 6, line 120: Since "z" is a parameter it should be given in italic type font.
- 4. Page 7, line 137: Please write "[...] inverting Eqs. (1) and (2)".
- 5. Page 7, line 142: It should read "[...] in the range of 2-20 µm [...]".
- 6. Page 7, line 143-144: I would suggest to write: "Care was taken to fill [...]".
- 7. Page 8, line 169: Please add "DOF" in the brackets behind "(depth of field)".
- 8. Fig. 14: I would suggest increasing the size of the symbols in the figures here.

We agree with the minor comments and applied the specific corrections according to the reviewer's suggestions.

**References**

Kashdan, J. T., Shrimpton, J. S., and Whybrew, A.: Two-Phase Flow Characterization by Automated Digital Image Analysis. Part 1: Fundamental Principles and Calibration of the Technique, Particle & Particle Systems Characterization, 20, 387–397, https://doi.org/10.1002/ppsc.200300897, 2003.

---

## Author Comment (AC3) · 17 Dec 2020

**Authors' Final Response**

Jakub L. Nowak, Moein Mohammadi, Szymon P. Malinowski

We are grateful to both the reviewers for the insightful comments and suggestions on our manuscript. We respond to them in detail below. Reviewers' comments are given in black, our anwers in blue. The responses mention specific corrections which were applied as the result of reviewers' suggestions. Next, we explain minor corrections introduced by ourselves. We also attach the full revised manuscript with all the changes marked (generated with latexdiff).

**Response to the Anonymous Referee #1**

**General comments**

It is challenging to characterize the sample volume of the shadowgraph system (Sect. 3) without accurate information regarding the properties of the plume (e.g. uniformity, size distribution of poly-dispersed droplets). Uncertainties in the calibration method can have large uncertainties on the droplet number concentration and size distribution. The assumptions and speculations in Sect. 3 are all justified, but some uncertainties remain. For example, the different lens systems show different dependencies of the number concentration on the horizontal, vertical and axial position (Fig. 8), some of which are attributed to the non-uniformity of the plume (x2, x4), whereas others are explained by instrumental flaws (x1). A better characterization of the plume would strengthen the interpretation.

We agree with the referee that the better characterization of the plume would strengthen the interpretation, i.e. allow to distinguish the effect of its non-uniformity from instrumental flaws. However, we were not able to do this with the available instrumentation. Therefore, we point out in the text that some particular conclusions cannot be delivered with high confidence based on our experiment.

The most important correction we developed involves limitation of the depth of field $z_{95}$ in Eq. (5). It was first derived analytically and the experiment proved to confirm its validity. It did not serve to derive the calibration, i.e. none of the coefficients used in data processing was fitted to the collected data. The calibration constants $a_1, \ldots, a_6$ were prior provided by the manufacturer and resulted from the calibration performed with the Patterson globe targets (Kashdan et al., 2003).

Concerning dependency of the number concentration on the position illustrated in Fig. 8, we do not claim that in case of lens setting x2 and x4 the effect is entirely due to the non-uniformity of the plume. What we mean is that such dependency might have been caused by the plume properties, hence we are not entitled to blame instrumental issues on the ground of our results. Lens setting x1 features such a large difference that, in our opinion, it is hardly possible to explain it with the properties of the plume only.

Regarding the particle sizing, a FMAG was used to produce mono-dispersed droplets (in the range of 15 to 72 μm in diameter). For the size calibration experiments, droplet diameters of 20.13, 39.35 and 57.55 μm were chosen. Would it be possible to produce size calibration experiments with smaller sizes (e.g. by using a Vibrating Orifice AerosolGenerator (VOAG); or PMMA/PSL spheres with atomizer or fluidized bed)? Previous studies have shown that sizing uncertainties are largest for the smallest particles, so I would recommend performing sizing experiments for smaller sizes and if possible include additional instruments for better validation. Furthermore, cloud droplets are generally smaller than 20 μm. For example, the cloud droplet size observed in Sect. 5 all lie below the smallest calibration size applied in this study (20.13 μm).

We are aware of this limitation. Naturally, it would be of advantage to verify the sizing of smaller droplets as we also expect largest uncertainties for the smallest particles. However, the instrument which was at our disposal (FMAG) can reach down to 17 μm in diameter using ultra purified water (UPW) as the fluid (Duan et al., 2016). In the course of the experiment, we tried to generate droplets smaller than 20 μm but this resulted in quite a wide spread of the diameters. In FMAG, the size is controlled by the liquid flow rate and the vibration frequency. To obtain small sizes, high vibration frequency is necessary (e.g. 200 Hz for the diameter of 17 μm, see Table 2 in Duan et al. (2016), which probably results in less-accurate breakage of the fluid stream into droplets and more likely collisions. The smallest diameter we used in this study (20.13 μm) was chosen as a compromise to avoid these effects.

We plan to develop a special calibration target composed of dots of known diameters, similar to that applied by Kashdan et al. (2003) but in the size range below 20 μm. Unlike the droplets, the position of the target in the sampling volume would be under control.

**Specific comments**

1. Page 1 line 19: You mention that researchers have to tackle intrinsic difficulties when using in situ and remote sensing observations. You could consider giving some examples describing the main challenges of in situ and remote sensing observations.

   We listed a few examples for each of the two approaches:

   > Researchers employing both strategies tackle with intrinsic difficulties. For instance, in-situ methods often face the dependence of the sample volume on particle size or air flow velocity, nonlinearity of Mie scattering intensity with respect to droplet size, aerodynamic effects related to the flow around or inside the instrument or aircraft, harsh conditions (incl. icing, wetting, temperature changes), necessity for handling large datasets or instantaneous data processing. Remote sensing provides the information with limited spatial resolution, hence microphysical properties represents only the average or integral over relatively large volumes which might be too simplistic to characterize inhomogeneous or multilayered cloud fields. On top of that, the retrievals are often dependent on the assumptions of specific size distribution or specific vertical structure of the atmosphere.

2. Page 2 line 44: Holographic systems have also been applied on balloon-borne platforms; e.g. HoloBalloon Ramelli et al. 2020, https://amt.copernicus.org/articles/13/925/2020/

We included this study in the short review of cloud droplet sizing instruments given in the introduction. We also updated the citation concerning the HALOHolo instrument. Together with Schlenczek et al. (2017) who actually performed ground-based measurements, we cited Schlenczek (2017) (PhD thesis) and Lloyd et al. (2020).

3. Page 2 line 48: Here you compare the sampling volume and frame rate of holographic and shadowgraph instruments. I would recommend to compare the sample volume rate as this is more meaningful (e.g. 15 cm3*5 fps = 75 cm3 s-1; 0.04 cm3*400fps = 16 cm3 s-1).

We agree, those values were mentioned in the revised text. In case of holography we used the value of 6 fps in this estimation as this is the true frame rate of e.g. HALOHolo and HoloGondel. Additionally, we cited the recent advancements with regard to the sample volume rate achieved in HoloBallon by Ramelli et al. (2020).

4. Page 2 line 50: You write that the shadowgraph technique did not gain common use in cloud measurements. This is misleading, as CPI is frequently used on aircrafts. Here are some examples:
Lawson et al. 2001: https://agupubs.onlinelibrary.wiley.com/doi/pdf/10.1029/2000JD900789
Lawson et al. 2010: https://agupubs.onlinelibrary.wiley.com/doi/pdf/10.1029/2009JD013017
Stith et al. 2002: https://journals.ametsoc.org/jamc/article/41/2/97/16088
Woods et al. 2018: https://agupubs.onlinelibrary.wiley.com/doi/pdf/10.1029/2017JD028068

We did not formulate the point clearly. What we actually meant is that shadowgraphy is not common in cloud droplet measurements in comparison to other methods. Additionally, we cited the first from the papers listed because it describes the capabilities of the CPI, and the last one to give the most recent specification on the size range measured by this instrument.

The corrected sentence reads:

Nevertheless, despite both its simplicity and many insightful laboratory experiments, e.g. concerning droplet collisions (Bordás et al., 2013; Bewley et al., 2013), shadowgraphy is not the first choice method in cloud droplet measurements.

5. Page 7 line 146: In Fig. 1 the flow was vertically aligned. Did you use a different flow direction for the experiments? Please specify.

Fig. 1. illustrates the components of the setup. The photograph was taken during another short test performed with the same instruments but different mounting of the pipe. It is neither analyzed nor discussed in the paper. During the measurements described in the paper, the outflow from the pipe was horizontally aligned. Unfortunately, no picture of this particular configuration is available.

Appropriate explanation was added to the text in sec. 3.1:

The flow velocity was estimated to be of the order of $10\,\mathrm{cm\,s^{-1}}$ and the direction of the flow was aligned horizontally from left to right (i.e. the direction of the pipe exit was perpendicular to what is shown in Fig. 1).

6. Page 9 Fig. 4: On page 7 line 142 you state that the poly-dispersed water droplets in the stream were in the size range 2-20 μm. Is there a way to verify this? Why do you observe a lot of droplets larger than 20 μm in Fig. 4?

Unfortunately, we do not have another droplet sizing instrument. The given approximate range comes from the works of P. Korczyk and others (Korczyk, 2008; Korczyk et al., 2012) who used the same source of droplets, i.e. an ultrasonic humidifier. They measured the size spectrum by collecting sedimenting droplets on a glass plate covered with an oil film to prevent evaporation and analyzing the microscopic photographs. Despite the same source, the delivery system and ambient thermodynamic conditions are somewhat different in our case, which might result in small changes in the actual droplet size distributions due to possible collisions and evaporation. Note also that we collected much larger statistics (i.e. total number of counts) than was feasible with their method, hence very infrequent large droplets are better represented in our dataset.

We added the explanation in the text:

> A dense stream of poly-disperse water droplets was generated with the use of an ultrasonic humidifier, the same as in the study of Korczyk et al. (2012) who measured the droplet size to be mostly in the range of 2-20 μm in diameter (Fig. 1 there). Differences in delivery method and in ambient conditions could result in a little different spectrum.

7. Page 14 line 280: I only see the gradual decrease from the center to the sides for x1 in the horizontal direction. It seems like you have more particles on the left side compared to the right side (maximum is not at 0 μm). Is this relating to the non-uniformity of the plume or to instrumental flaws?

Our interpretation of the logarithmic plot given in the text was too simplistic. We changed the quantity presented in panels (a), (b) of Fig. 8 by normalizing $N_V(x)$ or $N_V(y)$ by the mean concentration to show the relative changes. After this correction it is clear the maximum is located left from the center. We suppose this might be related to the non-uniformity of the plume as well as the instrumental flaws. Therefore, we cannot claim the specific instrumental issues based on the results.

The corrected description reads:

> Figure 8 in panels (a) and (b) shows normalized droplet concentration $N_V(x)/N_V$ and $N_V(y)/N_V$ suitably integrated over size and over other dimensions, divided by the total concentration in order to highlight the relative dependence on position inside the sample volume. The values mostly decrease gradually from the maximum (located left from the center) to the sides in case of horizontal direction and from the bottom to the top in case of vertical. The relative differences, except from very close to the edges, are of the order of 10 %. They are small enough to be possibly caused by the non-uniformities of the plume.

[Figure]

**Figure 8 corrected.** Dependence of droplet concentration on (a) horizontal, (b) vertical and (c) axial position.

8. Page 14 line 284: You speculate that the decrease in concentration in vertical direction for x1 is due to instrumental reasons (e.g. non-uniform illumination). Can you verify/quantify that based on the images? The shadow images for x1 in Fig. 12 don't show indications for non-uniform illumination. Please comment on that.

The VisiSize software has an option of background normalization. The user can select one image which serves as a background and all the further frames collected in the measurement are divided by it. Such an option was used in the detection experiment described in sec. 3 (Fig. 3). We speculate the background normalization ensures proper performance of the thresholds ($T_h$ and $T_p$) but does not substantially improve the signal-to-noise ratio in originally darker (weakly illuminated) regions. There, only a small part of the grayscale levels (256) is used.

We cannot verify/quantify this issue directly for our detection experiment because in the course of long runs, the images are not saved but instantly processed to derive particle statistics (see the list of them in Table 2). If one prefers to keep the captured images, the duration of an uninterrupted measurement is limited by the available computer RAM which in our particular configuration meant about 2000 frames (corresponding to about 66 seconds).

In the series shown in Fig. 12 some gradient of brightness is visible in case of lens setting x1, for instance the right edge seems to be slightly darker than the center and the left. The orientation of the gradient is then different than in the detection experiment (sec. 3) which might result from manual position adjustment. We recall our observation that the satisfactory adjustment for this lens setting is rather difficult to achieve. Importantly, most of the droplets measured during the sizing experiment (sec. 4) were close to the central part of the SV. Therefore, our findings are not affected by the illumination non-uniformity.

9. Page 14 line 292: You explain that the axial dependence is more difficult to evaluate. Fig. 8.c shows a sharp decrease of the concentration with increasing z-distance, which you attribute to the miscounting of the smaller droplets. Would it be possible to perform a similar experiment as in Sect. 4 where you generate a mono-dispersed particle distribution? Or are the concentrations produced by the FMAG too small?

The concentrations produced by the FMAG are indeed much smaller (0.3-3 mm$^{-3}$) in comparison to the dense plume in the first experiment. We speculate the exact value depends non-trivially on the settings of the FMAG and the conditions on the way from the nozzle exit to the sample volume of the shadowgraph (flow properties, evaporation, collisions). Moreover, the generated stream of droplets is quite localized which implies non-random probing of the sample volume, including the axial direction. The essential component of the detection experiment with the dense plume (sec. 3) was the close-to-uniform filling of the SV. In our opinion, appropriate study of detection properties involving FMAG-generated droplets would require precise control of the position and orientation of the droplet stream with respect to the focal plane of the shadowgraph which seems to be challenging.

10. Page 16 line 335: For the size calibration experiments, you produced droplet diameters of 20.13, 39.35 and 57.55 μm. Can you explain how these diameters were selected?

We intended to select 3 different sizes covering the range which is available with the particular FMAG generator at our disposal using UPW (17-69 μm, Duan et al. (2016)). As explained in our reply to the general comment above, the smallest diameter (20.13 μm) was chosen as a compromise in order to avoid spectrum broadening. Such effect is most probably related to the high vibration frequency which together with the liquid flow rate controls the output size.

We added a short explanation of the choice of the smallest size in the text:

> The smallest diameter was chosen to ensure a relatively narrow spectrum as we observed significant broadening for high vibration frequency which is necessary to generate yet smaller sizes. Presumably, this results in less accurate breakage of the fluid stream or more frequent collisions.

11. Page 18 line 353: You state that the left-side skewness of the tails in Fig. 11 implies partial evaporation. I only see a left-skewed tail for x2 and x1 for the diameters 39.25 and 57.55 μm, but not for x4 and the smaller size (20.13 μm). I would expect that the evaporation effect is largest for smaller particles, but this does not seem to be the case. How can you explain this pattern?

In our opinion, the observation made by the reviewer might be related to the size of the sample volume which increases with effective pixel size (decreases with magnification) as well as with the particle size (i.e. from the top-left to the bottom-right panel in Fig. 12). Other processes which were not controlled in our experiment might have also contributed to the observed results, e.g. ambient air properties, velocity of the droplets or some interactions among them.

Our speculation was explained in the text:

> Left-side skewness of the tails suggests partial evaporation. Although in general the effect of evaporation is expected to be more significant for small droplets, the skewness is evident for 39.25 and 57.55 μm measured with the lens settings x1 and x2. We speculate it might be related to the size of the sample volume which increases with the effective pixel size (decreases with magnification) as well as with the particle size (i.e. from the top-left to the bottom-right panel in Fig. 12). The position of the nozzle exit was adjusted so that the

center of the FMAG-generated droplet stream is as close as possible to the focal plane of the shadowgraph. We expect the droplets more distant from the central axis of the stream to be more likely partially evaporated because of the longer travel and exposure to the dry air blown from the area around the nozzle. Those can be detected in case of considerable sample volume but not in case of smaller SV. Importantly, this is only one of the effects which could have contributed to the observed result together with the ambient air properties, velocity of the droplets or some interactions among them.

12. Page 21 Fig. 14: Why do you have multiple times the same symbol in a, b and c? Are these different experiments?

   Yes, there were multiple measurement rounds performed with the same settings. Each symbol represents the mean diameter or the 1st-peak diameter obtained in a single measurement. The caption of the figure was updated accordingly.

13. Page 22 Fig. 15: Were other cloud probes deployed at the measurement site? On page 20 line 381 you say "After laboratory tests, the shadowgraph VisiSize D30 has been used for the first time to measure droplets in atmospheric clouds [...] to compare it with other probes already in service in cloud physics studies [...]". If other cloud probes were deployed at the same time, I would add the size distribution of additional probes in Fig. 15 for validation of the VisiSize D30 shadowgraph system. Alternatively, I would recommend performing a comparison campaign with other cloud probes in the future.

   During two observational periods in July and August 2019, the size distribution was measured also by the Phase Doppler Interferometer (PDI). The two instruments, intentionally located side-by-side were not simultaneously operational in all the runs but we have data from overlapping periods. This data is still subject to processing and detailed analysis. In case of the PDI, the sample volume depends non-trivially not only on droplet size but also on the wind velocity, highly variable in the turbulent experimental conditions. Therefore, we are working on proper systematic comparison which considers those issues and shows some limitations of ground-based measurements with the instruments developed for high true air speeds . We intend to describe the details and submit it to the AMT as soon as the analysis and interpretation is finished.

**Technical comments**

1. Page 5 line 93: I would suggest to write "[...] a method compensating for the effects[...]"

2. Page 6 Fig. 2: Consider choosing different colors for Th and Tp for better distinction.

3. Page 6 line 120: 'z' should be written in italic.

4. Page 8 line 165: 'z' should be written in italic.

5. Page 8 line 167: 'z' should be written in italic.

6. Page 15 line 308: capitalize 'D' in "2-dimensional (z, D) maps"

7. Page 19 Fig. 12: Consider adding different lines for single droplet/double collision/triple collision similar as in Figure 13 or at least of the FMAG diameter.

We agree with the technical comments and applied the specific corrections according to the reviewer's suggestions.

**Response to the Anonymous Referee #2**

**General comments**

Would it be possible to use the shadowgraph system to obtain some 3D information about the position of the cloud droplets?

Unfortunately, the diameter-dependent sample volume is generally too small to obtain the information about the 3D arrangement of the droplets in a typical atmospheric cloud. For instance, in Fig. 5. one can see the DOF for the droplet of a diameter 20 μm is about 200 μm (twice the $z_{max}$). Multiplied by the FOV (see Table 1) it results in the SV of about 5.90, 1.50 and 0.35 mm$^3$, for magnifications x1, x2 and x4, respectively. Even if the concentration was 1000 cm$^{-3}$ and all the droplets had 20 μm, we would expect on average around 5.9, 1.5 or 0.35 counts per frame, respectively. For smaller diameters, the DOF and hence the SV as well as the expected counts per frame are accordingly much smaller. For the field measurement described in sec. 5, the concentrations were estimated to about 928 cm$^{-3}$ and 798 cm$^{-3}$. During the first round with magnification x2, there were on average only 0.7 counts per frame. During the second round with magnification x4, there were on average only 0.08 counts per frame.

Another issue might be the shape of the SV resembling more a slice than a cube as the z-dimension is much smaller than x and y. On top of that, only the absolute value of $z$ coordinate of a droplet is estimated. It is unknown, on which side of the focal plane the object is.

Could you give any estimate for the largest detectable number concentration? In case of large number concentrations, for example, it might be possible that droplets "hide" one after the other.

Due to the small slice-shaped SV, the problem of the largest detectable concentration is not relevant for typical atmospheric clouds. Similarly to the previous question, we may consider mono-dispersed droplets of a diameter 20 μm and concentration 1000 cm$^{-3}$. Then, the expected number in the single cylinder of a diameter 40 μm and the height equal to the DOF (around 200 μm) is only $2.5 \cdot 10^{-4}$. For smaller droplets, it would be accordingly smaller. Therefore, the coincidence of two droplets at similar $(x, y)$ position seems highly improbable.

In the field measurements, there was on average even below 1 count per frame. The concentration in the plume used during the laboratory experiment was about two orders of magnitude larger which gave 18-54 counts per frame (depending on magnification). In the limited number of the inspected test images we did not spot coincidences. However, in the course of the longer

measurement the images are not stored but processed in real time (see sec. 2.1) which makes the problem of coincidences hardly possible to verify afterwards.

On the other hand, what we observed during the laboratory experiment is that the dense plume filling the considerable space between the camera and the laser but outside the SV can influence the average brightness of the image. The thresholds implemented in the software adapt to the average brightness but still such shadow diminishes signal-to-noise ratio of individual particles as discussed for the magnification x1 in sec. 3.5. In our opinion, the effect is not significant in atmospheric clouds due to smaller concentration, hence limited scattering.

Concerning the application of the shadowgraph system in the atmospheric cloud (Section 5): Has there been any other cloud probe which measured the size distribution at the same time as the shadowgraph system? If so, could you present a comparison of the obtained droplet size distributions and its moments, respectively? If not, I would suggest, for a future study, to have a comparison to another cloud probe.

During two observational periods in July and August 2019, the size distribution was measured also by the Phase Doppler Interferometer (PDI). The two instruments, intentionally located side-by-side were not simultaneously operational in all the runs but we have data from overlapping periods. This data is still subject to processing and detailed analysis. In case of the PDI, the sample volume depends non-trivially not only on droplet size but also on the wind velocity, highly variable in the turbulent experimental conditions. Therefore, we are working on proper systematic comparison which considers those issues and shows some limitations of ground-base measurements with the instruments developed for high true air speeds. We intend to describe the details and submit it to the AMT as soon as the analysis and interpretation is finished.

**Specific comments**

1. Page 7, line 145-146: Looking on Fig. 1, I would think that the direction of the droplet flow was vertically aligned.

    Fig. 1. illustrates the components of the setup. The photograph was taken during another short test performed with the same instruments but different mounting of the pipe. It is neither analyzed nor discussed in the paper. During the measurements described in the paper, the outflow from the pipe was horizontally aligned. Unfortunately, no picture of this particular configuration is available.

    Appropriate explanation was added to the text in sec. 3.1:

    > The flow velocity was estimated to be of the order of $10 \ \mathrm{cm \, s^{-1}}$ and the direction of the flow was aligned horizontally from left to right (i.e. the direction of the pipe exit was perpendicular to what is shown in Fig. 1).

2. Fig. 5c: For diameters larger than 30 µm, $z_{max}$ deviates significantly from the analytical approximation $z_{95}$. Do you have any explanation for that?

    The discrepancy relates to the poor statistics of large droplets. Each point in the plot represents one diameter bin and shows the distance from the focal plane corresponding to the furthest detected droplet from that bin. Diameters larger

than 30 µm are very scarce in the plume (see the size distribution in Fig. 7), so the number of detections is also small and they do not fill equally the entire sample volume. Therefore, it can happen that all of them randomly appear at the positions closer to the focal plane than the maximum possible distance $z_{95}$. For instance, in the case of the measurement with the magnification x4 presented in Fig. 5c, there were only 3 counts in the last bin and only 15 in the one before last. The sample volume decreases strongly with magnification implying the limited chance of detecting infrequent large droplets which explains why the effect is most pronounced for x4.

We added a short comment to the text:

> The discrepancy observed for largest sizes is related to the poor statistics, i.e. small number of counts of large droplets which are infrequent in the measured plume.

3. Fig. 7 and Fig. 15 and the corresponding text: In Fig. 7 only DSDs applying methods"def" and "ind" are shown, for given reasons. However, later in Fig. 15, DSDs for "def","cor" and "ind" are shown. Wouldn't it make more sense, to show DSDs for methods"def", "cor" and "ind" in Fig. 7 and explain that there are only small differences between methods "cor" and "ind" and then only show DSDs for "def" and "ind" in Fig. 15?

We added the missing method to Fig. 7 and decided to keep all of them in Fig. 15 for consistency. The caption of Fig. 7 was modified accordingly.

4. Page 14, line 280: This gradual decrease from the center to the sides is obvious for x1, but not really for x2 and x4. Could you please comment on that?

Our interpretation of the logarithmic plot given in the text was too simplistic. We changed the quantity presented in panels (a), (b) of Fig. 8 by normalizing $N_V(x)$ or $N_V(y)$ by the mean concentration to show the relative changes. After this correction it is clear the maximum is located left from the center.

[Figure]

**Figure 8 corrected.** Dependence of droplet concentration on (a) horizontal, (b) vertical and (c) axial position.

The corrected description reads:

Figure 8 in panels (a) and (b) shows normalized droplet concentration $N_V(x)/N_V$ and $N_V(y)/N_V$ suitably integrated over size and over other dimensions, divided by the total concentration in order to highlight the relative dependence on position inside the sample volume. The values mostly decrease gradually from the maximum (located left from the center) to the sides in case of horizontal direction and from the bottom to the top in case of vertical. The relative differences, except from very close to the edges, are of the order of 10 %. They are small enough to be possibly caused by the non-uniformities of the plume.

5. Page 14, line 284-291: I think the speculation given here is reasonable. However, could this also be true for x2 and x4? But here more (smaller) droplets are detected, both in a smaller FOV compared to x1 which compensates this feature?

As far as we understand, the reviewer suggests the suspected y-dependent concentration concerns large droplets regardless of the magnification setting used. This effect is then supposedly not clearly visible for x2 and x4 because of the dominant number of small droplets detected in a more uniform manner.

We verified this claim to be true. The concentration $N_V(y)$ of the droplets larger than 12 μm only is indeed more y-dependent than the same quantity for the droplets smaller than 12 μm, in case of both magnifications x2 and x4. We cannot exclude the contribution of gravity sorting to that effect. However, the relative decrease of the $N_V(y)$ of large droplets from the bottom to the top is significantly smaller than over the same distance in case of the magnification x1.

The exact influence of the non-uniform illumination on the results of DSD and DNC is hardly possible to be estimated with the available data. We emphasize the importance of precise alignment of the laser and camera lens positions. This is done manually, so the positioning can differ slightly between the experiments. As stated in the paper, achieving close to uniform illumination is much easier for magnifications x2 and x4 than x1 because of the smaller FOV to be fitted into the same laser beam.

We modified the text:

> The y-dependence of the concentration is more pronounced for large droplets above 12 μm (not shown) which might be the influence of gravity sorting. However, in case of the series recorded with lens setting x1, the concentration falls with height y by a factor of more than 10 from the bottom to the top of the FOV. We speculate this effect is of instrumental origin, as the difference is rather too large (almost exponential) and the timescale too short to allow for the explanation only by gravity sorting of the droplets in the plume.

6. Page 15, Line 318: Could you please provide a size range where you would not use x1. As later said, x1 makes sense for larger droplets in the drizzle and rain size range. Please make clearer under which circumstances you would avoid using x1, and vice versa.

We estimated the minimum diameter for uniform detection to 12 μm so the magnification x1 is not recommended to measure any droplets smaller than that. Another limit, related to the issue of illumination, is not so precise because the image quality depends to some extent on the accuracy of manual alignment. In order to avoid poor signal-to-noise ratio

we would suggest, as a rule of thumb, a diameter an order of magnitude larger than the effective pixel size which results in around 40 µm. On the other hand, the objects need to fit into the FOV, so the largest measured size should be a few times smaller than the shorter dimension (4.1 mm). Eventually, we expect the setting x1 to perform well in the range of roughly 40-400 µm. So far, we have not collected much data on drizzle to confirm this expectation experimentally. Nevertheless, in the sizing experiment (sec. 4) we had indeed no issues for 39.35 µm droplets but still some inaccuracies for 20.13 µm.

We included this information in the text:

> As a consequence, the authors discourage using lens setting x1 in further studies of cloud droplets. The large minimum particle size for uniform detection reported earlier ($\sim$12 µm) makes this option of limited utility anyway. Instead, x1 can possibly serve well for the measurements of drizzle drops. In order to avoid the illumination and signal-to-noise issue described above, we would suggest, as a rule of thumb, the lower limit an order of magnitude larger than the effective pixel size (3.69 µm). On the other hand, the objects need to fit into the FOV, so the largest measured size should be a few times smaller than the shorter dimension (4.1 mm) which eventually results in the conservative range of roughly 40-400 µm. Notwithstanding, so far insufficient data on drizzle has been collected to confirm this expectation experimentally.

7. Fig. 9 and respective text: Does the discrepancy in the first and second z-bin has any consequences on the calculation of the DSD and DNC?

We consider the question relevant however cannot estimate the effect quantitatively based on the available data. In order to investigate this issue we increased the number of $z$-bins from 20 to 80 which implied the changes in Fig. 8c and Fig. 9. This operation allowed to identify a lower limit on the distance $z$ and the probable reason for the discrepancy observed earlier - diffraction around the objects close to the focal plane. In the paragraphs added to the text we discuss the mechanism and speculate the influence on the DSD and the DNC within the valid size range is rather minor. In our opinion, one solution to systematically eliminate the problem would be to add the term representing diffraction to Eq. (1). This requires repeating the calibration procedure to obtain the correct new values of $a_1$ and $a_2$. At best, also implementing the corrected equation in the software processing routine.

We corrected the interpretation of Fig. 8c:

> Interestingly, the first $z$ ranges contain a much smaller number of droplets than the maximum located further from the focal plane, regardless of the lens setting.

We added the paragraphs discussing the issue:

> On the other side, the minimum distance $z_{min}$ for a given diameter is well approximated by assuming the halo area in the focal plane equal to the diffraction term ($A_h = \pi D a_5$), analogously to Fig. 4, and solving Eq. (1) for $z$ (blue line).

[Figure]

**Figure 9 corrected.** Variability of droplet detection properties in size-space domain for lens setting (a) x1, (b) x2, (c) x4.

(...)

In experimental runs with higher magnifications, i.e. lens setting x2 and x4, the $(z, D)$ map features the decrease of concentration with $z$ from the maximum located a bit above $z_{min}$. The same effect was noticed in Fig. 8 panel (c). Most probably, it relates to the diffraction which is not included in the modeled dependence of the halo area $A_h$ on the diameter $D$ and distance $z$ in Eq. (1). Namely, the equation is not correct in the limit of small $z$ because it implies the object in the focal plane ($z = 0$) should be ideally sharp ($A_h = 0$). Consequently, in this limit the calculated $z$ position is overestimated with respect to the true one. The counts representing droplets standing very close to the focal plane are shifted to the further $z$-bins in Fig. 9. The extent of this shift is probably not constant but decreases with the true $z$. Hence, the counts cumulate at some point above $z_{min}$ creating a maximum in $N_V(z)$. We expect the shift to decrease because the calibration constants $a_1$ and $a_2$ were fitted by the manufacturer in the procedure resembling Kashdan et al. (2003) so that the Eq. (1) performs satisfactorily in the range of defocus distances and particle diameters typical for industrious applications, i.e. a bit larger than analysed here. Therefore, the estimated $z$ should approach the true one with increasing defocus and droplet diameter.

The shift of the estimated $z$ positions with respect to the true ones is most pronounced in case of the small distances from the focal plane. Importantly, it should have no effect on the accuracy of the sample volume calculation, hence the DNC and the DSD, as long as the $z_{95}$ is not overestimated but represents the true distance at which the droplets are no longer counted. We expect this condition to be met if the largest possible defocus $z_{95}$ is significantly higher than the smallest $z_{min}$. For instance, they differ by a factor of two for diameters larger than 8.0, 6.3, 4.8 µm in case of lens settings x1, x2 and x4, respectively. Those values are close to the minimum diameter for uniform detection estimated in sec. 3.4. It implies the influence on the DSD and the DNC within the valid size range is probably minor. However, we cannot quantify it with high confidence based on the available data.

**Minor comments**

1. Page 2, Line 50: I would suggest to write either "shadowgraphy" or "the shadowgraph technique".

2. Page 3, line 70: Please add "the" at the beginning of the sentence: "The two main parts[...]".

3. Page 6, line 120: Since "z" is a parameter it should be given in italic type font.

4. Page 7, line 137: Please write "[...] inverting Eqs. (1) and (2)".

5. Page 7, line 142: It should read "[...] in the range of 2-20 µm [...]".

6. Page 7, line 143-144: I would suggest to write: "Care was taken to fill [...]".

7. Page 8, line 169: Please add "DOF" in the brackets behind "(depth of field)".

8. Fig. 14: I would suggest increasing the size of the symbols in the figures here.

We agree with the minor comments and applied the specific corrections according to the reviewer's suggestions.

**Additional corrections**

While revising the manuscript, we found a mistake in Eqs. (6), (7) and (8). As the text explains, the effective cross-sectional area of camera sensor, FOV, is reduced by the margin of the width equal to a diameter under consideration. In the equations, we wrongly used $2D$ instead of $D$, confusing diameter with radius. This term was corrected. The change concerns the equations in the manuscript and the captions in Fig. 6. All the calculations were performed correctly.

[revised manuscript text omitted]

---

## Author Response (AR2)

**Response to the editor's comment**

We are grateful to the editor for accepting our paper for publication in the journal and for handling the review process. The technical correction concerning the citations of Schlenczek's papers was applied as requested.